# THE ANATOMY OF ALIGNMENT: DECOMPOSING PREFERENCE OPTIMIZATION BY STEERING SPARSE FEATURES

## ABSTRACT

Prevailing alignment methods induce opaque parameter changes, obscuring what models truly learn. To address this, we introduce Feature Steering with Reinforcement Learning (FSRL), a framework that trains a lightweight adapter to steer model behavior by modulating interpretable sparse features. First, we theoretically demonstrate that this mechanism is expressive enough to approximate the behavioral shifts of post-training processes. We then apply FSRL to preference optimization and perform a causal analysis of the learned policy. Our analysis reveals a crucial insight: the model learns to reward stylistic presentation as a proxy for quality, disproportionately relying on features related to style and formatting over those tied to alignment concepts like honesty. By effectively optimizing the preference objective, FSRL serves as a transparent proxy for observing the alignment process. Overall, FSRL offers an interpretable control interface and a practical way to diagnose how preference optimization pressures manifest at the feature level.

## 1 INTRODUCTION

Large Language Models (LLMs) are typically aligned with human preferences through post-training methods like Reinforcement Learning from Human Feedback (RLHF) (Ouyang et al., 2022). This fine-tuning induces parameter updates across the model's underlying weights. Consequently, the newly learned alignment behaviors and the model's original capabilities are encoded in the same parameters, making them difficult to disentangle. When models trained with RLHF subsequently exhibit undesirable behaviors like sycophancy or reward hacking (Perez et al., 2023; Shah et al., 2022), identifying their root cause becomes challenging. This opacity motivates the need for tools that can decompose the alignment process into transparent, auditable components.

Mechanistic interpretability offers a way to make alignment more transparent by exposing and manipulating a model's internal concepts. At its core is the *Linear Representation Hypothesis*, which suggests that high-level concepts correspond to linear directions in activation space (Elhage et al., 2022). Sparse Autoencoders (SAEs) provide a practical method for uncovering these directions by decomposing dense activations into a sparse basis of largely monosemantic features (Huben et al., 2024; Rajamanoharan et al., 2024). These features capture diverse phenomena, ranging from "code syntax" to "flattery", and can often be assigned interpretable labels using automated methods (Huben et al., 2024; Bills et al., 2023; Paulo et al., 2025). The resulting feature vocabulary enables not only analysis of what models represent, but also a potential interface for directly steering their behavior.

Building on this foundation, we propose **Feature Steering with Reinforcement Learning (FSRL)**, a framework that uses the interpretable feature vocabulary in SAEs as a direct interface for alignment. Conceptually, FSRL acts as a 'Feature Adapter'- combining the dynamic, input-dependent control of parameter-efficient fine-tuning with the transparency of feature steering. Instead of fine-tuning the entire model, FSRL operates on a frozen LLM together with its SAE, and trains a lightweight adapter with reinforcement learning to learn a policy for modulating SAE features, as illustrated in Figure 1. This design keeps the model's underlying capabilities intact in the frozen LLM, while channeling the learned alignment behavior through steering interpretable SAE features.

**Contributions** In this work, we introduce Feature Steering with Reinforcement Learning (FSRL), a framework that aligns a frozen LLM by training a lightweight adapter to steer its interpretable SAE features. We first establish the soundness of this approach by theoretically demonstrating that FSRL's activation-space corrections are functionally equivalent to a class of LoRA updates. Empirically, FSRL effectively optimizes the preference objective on UltraFeedback, though we find this optimization degrades generation coherence. We then leverage FSRL's transparency to perform a causal analysis of the learned policy. This analysis reveals a crucial insight: the model learns to reward stylistic presentation as a proxy for quality, disproportionately relying on features related to style over those tied to alignment concepts like honesty. Finally, we validate this mechanism by ablating style features, showing that this surgical intervention partially restores generation quality. These findings establish FSRL as a general method for diagnosing how alignment pressures manifest at the feature level.

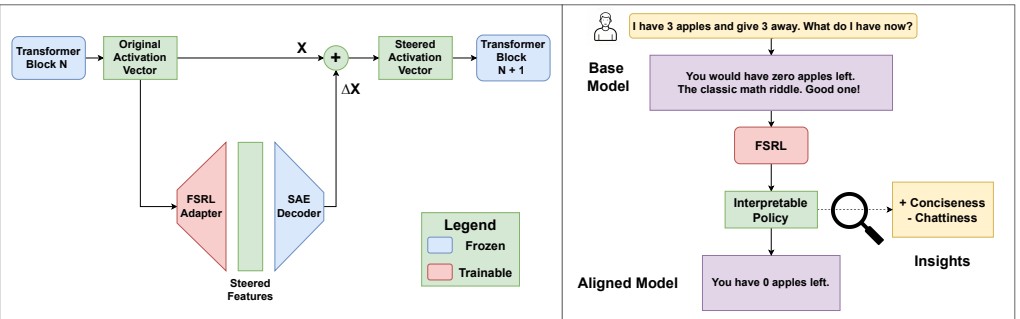

Figure 1: **The FSRL Framework for Interpretable Alignment. (a) FSRL Architecture:** At a given layer, the original activation vector is processed by a trainable adapter. The adapter outputs a sparse vector of steered features, which are transformed by a frozen SAE decoder into a correction vector. This correction is added to the original activation to steer the model's behavior. **(b) Application for Mechanistic Insight:** FSRL replaces opaque alignment processes with a transparent one by learning a policy over a basis of interpretable, monosemantic SAE features. This allows the learned alignment pressures to be decomposed into concrete actions on meaningful concepts.

## 2 BACKGROUND

We build on three key components: Sparse Autoencoders (SAEs) for creating an interpretable interface, Simple Preference Optimization (SimPO) to optimize a policy on a preference dataset, and a large annotated dataset to train our system.

**Sparse Autoencoders (SAEs)** SAEs are an unsupervised method for representing model activations as a sparse set of interpretable features (Huben et al., 2024; Anthropic, 2023). Each SAE consists of an encoder and a decoder. Given a model's hidden activation $\mathbf{x} \in \mathbb{R}^d$, the encoder first maps it into a higher-dimensional feature vector $\mathbf{f} \in \mathbb{R}^{d_{\text{sae}}}$ with $d_{\text{sae}} > d$:

$$\mathbf{f} = \text{ReLU}(W_{\text{enc}}\mathbf{x} + \mathbf{b}_{\text{enc}}), \tag{1}$$

where $W_{\text{enc}} \in \mathbb{R}^{d_{\text{sae}} \times d}$ and $\mathbf{b}_{\text{enc}}$ are encoder parameters. The decoder then reconstructs the original activation from $\mathbf{f}$:

$$\hat{\mathbf{x}} = W_{\text{dec}}\mathbf{f} + \mathbf{b}_{\text{dec}}, \tag{2}$$

where $W_{\text{dec}} \in \mathbb{R}^{d \times d_{\text{sae}}}$ and $\mathbf{b}_{\text{dec}}$ are decoder parameters. The columns of $W_{\text{dec}}$ form a dictionary of learned feature vectors. In particular, SAEs are trained such that each activation can be decomposed into only a few features, achieved by adding an $\ell_1$ penalty to the reconstruction loss. The total loss function is therefore:

$$\mathcal{L}(\mathbf{x}) = \|\mathbf{x} - \hat{\mathbf{x}}\|_2^2 + \alpha\|\mathbf{f}\|_1, \tag{3}$$

where $\alpha$ is a hyperparameter that controls the trade-off between reconstruction fidelity and feature sparsity. While this formulation is common, other SAE variants achieve sparsity through different mechanisms, such as the JumpReLU activation function (Rajamanoharan et al., 2024) or the Top-K operator (Bussmann et al., 2024).

SAE features can also be used for intervention. As each feature corresponds to a direction given by a column of $W_{\text{dec}}$, modifying an activation $\mathbf{x}$ by $\mathbf{x}' = \mathbf{x} + \lambda W_{\text{dec}}^{(i)}$ can steer the model's behavior in predictable ways. This property, known as *feature steering*, highlights that SAEs features are not only descriptive, but can also be used as actionable controls on model behavior.

**Simple Preference Optimization (SimPO)**  SimPO is an efficient algorithm for aligning language models with human preferences (Meng et al., 2024). It operates directly on a dataset $\mathcal{D}$ of preference triplets $(x, y_w, y_l)$, where $x$ is a prompt, $y_w$ is the preferred (chosen) response, and $y_l$ is the less preferred (rejected) response.

The objective is a modified Bradley-Terry loss with a target reward margin $\gamma$, which encourages the model to confidently separate $y_w$ and $y_l$:

$$\mathcal{L}_{\text{SimPO}}(\pi_\theta) = -\mathbb{E}_{(x,y_w,y_l)\sim\mathcal{D}} \left[ \log \sigma \left( \frac{\beta}{|y_w|} \log \pi_\theta(y_w|x) - \frac{\beta}{|y_l|} \log \pi_\theta(y_l|x) - \gamma \right) \right], \quad (4)$$

where $\beta$ is the temperature/scaling parameter, $|y|$ the sequence length and $\sigma(\cdot)$ the sigmoid function.

We adopt SimPO for its ability to match the performance of Direct Preference Optimization (DPO) (Rafailov et al., 2024) without requiring a separate reference model. This makes it possible to efficiently train the model (or FSRL adapter) directly on a preference dataset.

**Preference Dataset**  In this work, we use the UltraFeedback dataset (Cui et al., 2024). Specifically, we utilize the version of this dataset annotated with the Absolute-Rating Multi-Objective Reward Model framework (Wang et al., 2024). Our choice of this dataset is motivated by its use in the SimPO paper, which allows for a direct comparison, isolating the impact of our proposed FSRL framework rather than confounding it with dataset variations.

## 3  METHODOLOGY

We present Feature Steering with Reinforcement Learning (FSRL), a framework for transparently aligning LLMs by training a policy to steer sparse SAE features of a frozen model. In this section, we describe the system architecture, the training procedure, and the experimental configuration used for evaluation.

### 3.1  SYSTEM ARCHITECTURE

FSRL intervenes at a single chosen layer of a frozen LLM by steering the residual stream with a sparse, learned set of feature directions (Figure 1). At this layer, the residual activation $\mathbf{x} \in \mathbb{R}^d$ is first translated by the SAE into a sparse feature vector $\mathbf{f} \in \mathbb{R}^{d_{\text{sae}}}$. To decide how these features should be modulated, the same $\mathbf{x}$ is also given to a trainable adapter $\pi_\phi$, which outputs a sparse steering vector $\mathbf{v} \in \mathbb{R}^{d_{\text{sae}}}$. In effect, $\pi_\phi$ learns both the subset of features to target, as well as the direction and magnitude in which to steer them.

**Adapter Implementation**  We implement the adapter as a single feedforward layer with parameters $\phi = (W_a, \mathbf{b}_a, \boldsymbol{\tau})$, where $W_a \in \mathbb{R}^{d_{\text{sae}} \times d}$, $\mathbf{b}_a \in \mathbb{R}^{d_{\text{sae}}}$, and $\boldsymbol{\tau} \in \mathbb{R}_+^{d_{\text{sae}}}$ is a vector of learnable positive thresholds. Its output is produced by a coordinate-wise soft-thresholding activation function:

$$\mathbf{v} = \pi_\phi(\mathbf{x}) = \text{sign}(W_a\mathbf{x} + \mathbf{b}_a)\text{ReLU}(|W_a\mathbf{x} + \mathbf{b}_a| - \boldsymbol{\tau}). \quad (5)$$

We adapt this activation function from learned approximations of sparse coding (Gregor and Le-Cun, 2010). Unlike a standard ReLU, this function enables a tri-state intervention that improves interpretability: positive values amplify a feature, negative values suppress it, and values in the dead zone between $-\tau_i$ and $+\tau_i$ leave the feature unchanged. We validated this choice through architectural ablations detailed in Appendix F, which confirm that the ability to both amplify and suppress

features leads to a significantly sparser and more effective policy than an amplification-only ReLU approach..

**Applying Steering** The steering vector $\mathbf{v}$ specifies how SAE features are modulated. We obtained the steered activation by adding the decoded steering adjustment back into the residual stream:

$$\mathbf{x}_{\text{steered}} = \mathbf{x} + \text{Decoder}(\mathbf{v}). \tag{6}$$

Hence, given the input activation, the adapter learns to output a steering vector $\mathbf{v}$ that steers the model's output to be better aligned with the preference objective. In practice, we implement the update using a reconstruction-error variant (see Appendix A).

We favored this learned, dynamic approach over static heuristics. We empirically demonstrate that static steering vectors fail to adequately minimize the preference loss compared to our dynamic adapter (see Appendix H). Furthermore, we find that our learned sparsity policy is significantly more efficient and sparser than fixed top-k budgets (see Appendix G). Beyond these performance benefits, a trainable adapter allows the system to be optimized against any differentiable objective, ensuring FSRL is flexible enough for applications beyond preference optimization.

### 3.2 THEORETICAL JUSTIFICATION

While FSRL can align models with the training objective in practice, it is important to establish why its restricted form of intervention should, in principle, be expressive enough to match other fine-tuning methods. To this end, our theoretical justification shows that FSRL is a principled approach by demonstrating its functional equivalence to a restricted, yet powerful, class of low-rank adaptation (LoRA) updates (Hu et al., 2021). While FSRL's practical effectiveness is contingent on the capacity of its underlying SAE, our theory shows that its adaptation mechanism is sound.

The core of our proof, detailed in Appendix B, is that FSRL's activation-space corrections are functionally equivalent to a class of input-dependent LoRA updates. The FSRL update, $\mathbf{x}_{\text{steered}} = \mathbf{x} + \Delta(\mathbf{x})$, injects an additive correction into the residual stream. When passed to a downstream linear layer, this is algebraically equivalent to applying an effective weight update, $\Delta W[\mathbf{x}]$, whose rank is dynamically determined by the number of actively steered SAE features.

This equivalence is significant because it connects FSRL to the established foundations of LoRA. Recent work by Zeng and Lee (2024) proved that LoRA possesses sufficient expressive power to match a target model, given enough rank. While FSRL inherits these guarantees in principle, our single-layer intervention is a constrained application of this theory. Specifically, the adapter's policy is conditioned only on the activation at one layer, meaning it cannot distinguish between different upstream computational paths that yield the same activation vector. Despite this limitation, the connection confirms FSRL as a valid optimization method. Crucially, because FSRL is constrained to express its policy through the SAE's interpretable basis, the policy it learns provides a robust and transparent reflection of the optimization pressures driving the alignment task.

### 3.3 TRAINING CONFIGURATION

The adapter's parameters are optimized using the SimPO algorithm (Meng et al., 2024). To encourage a sparse and interpretable policy, we augment the training objective with an $\ell_1$ penalty on the steering vector, controlled by a coefficient $\alpha$. In addition to this proxy-based sparsity, we also investigated a more direct method using a JumpReLU activation (Rajamanoharan et al., 2024) in the adapter to directly optimize the $\ell_0$ norm. However, this proved to be difficult to tune within our framework (see Appendix E).

We evaluate our approach on both the **Gemma-2-2B-it** and **Gemma-2-9B-it** models (Gemma Team, 2024) using pre-trained SAEs from GemmaScope (Lieberum et al., 2024). For training, we use the **UltraFeedback dataset** (Cui et al., 2024). Our primary experimental decisions involved selecting the intervention layer and the sparsity coefficient. We performed a sweep across transformer layers and $\alpha$ values for both models to identify configurations that balanced steering vector sparsity with SimPO validation loss. We independently validated this layer selection using a computationally cheaper linear probing heuristic (see Appendix C). Detailed methodology for these sweeps and the final hyperparameters for both models are provided in Appendix D.

## 3.4 COMPARATIVE EVALUATION

To contextualize the performance of our FSRL-steered models, we establish baselines for comparison. For the 2B scale, we trained our own baseline consisting of the same instruction-tuned model fully fine-tuned using the standard SimPO algorithm. For the 9B scale, to ensure a rigorous comparison against the state-of-the-art and eliminate potential errors from our own training setup, we utilize the official public model checkpoint provided by the SimPO authors. The training configuration for our 2B baseline mirrors that of our FSRL adapter where applicable, with a decrement in the learning rate to ensure stable convergence (see Appendix D).

## 4 VALIDATING THE ALIGNMENT POLICY

We emphasize that FSRL is designed as a diagnostic tool rather than a competitor to full fine-tuning. Therefore, we benchmark the models primarily to verify that the adapter successfully captures the optimization signal. We compare performance against the base models and their fully fine-tuned SimPO counterparts, which serve as the non-interpretable performance ceiling.

We assess performance on MMLU (Hendrycks et al., 2021) for general knowledge, TruthfulQA (Lin et al., 2022) for truthfulness, and GSM8K (Cobbe et al., 2021) for mathematical reasoning. Evaluations were performed using the Language Model Evaluation Harness (Gao et al., 2024). The results are presented in Table 1.

Table 1: Benchmark performance for Gemma-2-2B-it and Gemma-2-9B-it models. FSRL optimizes the preference objective across model scales. Bold values indicate the best performance on a given metric within each model size group. We denote TruthfulQA as TQA for brevity

| Model | | MMLU ↑ | TQA (MC2) ↑ | GSM8K ↑ | Loss ↓ |
|---|---|---|---|---|---|
| | Baseline | 30.11 | 55.77 | **53.45** | 6.99 |
| Gemma-2-2B-it | SimPO Full | **50.28** | **61.35** | 4.40 | **2.19** |
| | FSRL | 41.95 | 56.10 | 7.05 | 2.58 |
| | Baseline | 33.86 | 61.02 | 75.73 | 6.09 |
| Gemma-2-9B-it | SimPO Full[1] | **58.24** | 59.4 | **77.78** | 2.74 |
| | FSRL | 43.69 | **62.08** | 0.00 | **2.46** |

Our results confirm that FSRL effectively optimizes the preference objective. Despite the theoretical constraints of a single-layer intervention discussed in Section 3.2, the adapter successfully minimizes preference loss across model scales. The 2B model illustrates a distinct trade-off: it preserves more mathematical reasoning capabilities than the full fine-tune, though it lags in the other benchmarks. The dynamic shifts at the 9B scale. Here, FSRL achieves the lowest preference loss and the highest TruthfulQA score, surpassing even the fully fine-tuned baseline. This optimization comes at the cost of a collapse in mathematical reasoning. We hypothesize this stems from the entanglement of concepts within the SAE; the features necessary for preference optimization may be closely linked to those required for mathematical ability, causing the adapter to disrupt reasoning capabilities when optimizing for preferences despite the enforcement of a sparsity penalty.

## 5 MECHANISTIC INSIGHTS INTO THE ALIGNMENT PROCESS

Having established that FSRL successfully captures the optimization signal, we now leverage its primary advantage: interpretability. To analyze the policy at a conceptual level, we developed an automated pipeline to classify SAE features based on their text-based explanations. We focus on two categories: **alignment features**, which encompass abstract concepts such as ethics, safety, and honesty; and **style features**, which relate to structural presentation elements like markdown syntax, list formatting, and punctuation. This automated process was validated against manual annotations, achieving reliable agreement with MCC scores ranging from 0.448 to 0.764 (details in Appendix J).

---

[1]We use the model provided by Meng et al. (2024) on HuggingFace: princeton-nlp/gemma-2-9b-it-SimPO

**Examining Feature Activations**  To understand how the adapter uses different types of features, we examine the composition of its feature activations. The FSRL adapter outputs a steering vector with an average $\ell_0$ norm of 95 for the 2B model and 58 for the 9B model (compared to the SAE baselines of 73 and 130). The 9B adapter is significantly sparser than its underlying SAE, while the 2B adapter is slightly denser. Given these distinct shifts in density, a simple raw count of active features can be misleading. We therefore analyze the proportion of active features belonging to a given category at each token, relative to the base SAE's activation patterns.

We measured this composition using activations derived from the preference dataset. As summarized in Table 2, this analysis reveals a consistent strategy across scales. For both the 2B and 9B models, the adapter learns to significantly decrease the proportional activation of alignment features (by $\sim$43% and $\sim$54% respectively) while simultaneously and substantially increasing the proportional activation of style features (by $\sim$150% and $\sim$256% respectively). This opposing pattern suggests the learned policy applies a general strategy of suppressing abstract alignment concepts in favor of amplifying stylistic ones. However, activation frequency does not imply utility. We therefore employ causal analysis to determine which of these actions drives optimization performance.

Table 2: Aggregate steering effect on the composition of active features for 2B and 9B models. 'SAE Baseline' is the average proportion of active features in a category for the unmodified model. 'Relative Change' is the percent change in this proportion caused by the FSRL adapter.

| Model | Feature Type | SAE Baseline (%) | Relative Change (%) |
|---|---|---|---|
| Gemma-2-2B-it | Alignment | 22.83 | -43.52 |
| | Style | 19.43 | 154.82 |
| Gemma-2-9B-it | Alignment | 19.21 | -54.19 |
| | Style | 11.71 | 256.48 |

**Intervening on Feature Activations**  For each category, we disabled the adapter's intervention by setting the corresponding components of its output steering vector to zero. We measured the impact of this ablation directly on the SimPO loss. Using the training objective as the metric allows us to make direct claims about the optimization process itself, revealing which feature categories are responsible for minimizing the preference loss, rather than observing indirect effects on downstream benchmarks. A null hypothesis where all features contribute equally would predict that the loss increases in proportion to the number of features ablated. Our results in Table 3 deviate sharply from this expectation.

Table 3: Causal contribution of feature categories for 2B and 9B models. 'Features Ablated' is the total number of features in a category. 'Loss per Feature' normalizes the resulting increase in SimPO loss by this count.

| Model | Ablation Condition | Features Ablated | SimPO Loss ↓ | Loss per Feature |
|---|---|---|---|---|
| Gemma-2-2B-it | None (Full Steering) | 0 | 2.58 | – |
| | Alignment Features | 11,143 | 2.63 | $4.49 \times 10^{-6}$ |
| | Style Features | 15,391 | 5.12 | $1.65 \times 10^{-4}$ |
| | Both Categories | 26,534 | 5.45 | $1.08 \times 10^{-4}$ |
| Gemma-2-9B-it | None (Full Steering) | 0 | 2.46 | – |
| | Alignment Features | 2,920 | 2.67 | $7.19 \times 10^{-5}$ |
| | Style Features | 1,889 | 3.21 | $3.97 \times 10^{-4}$ |
| | Both Categories | 4,807 | 4.10 | $3.41 \times 10^{-4}$ |

The Loss per Feature column quantifies the disproportionate impact of each category. For the 2B model, the average loss increase per style feature is nearly 37 times greater than that of an alignment feature. For the 9B model, while the gap narrows, style features still exhibit a causal impact nearly 6 times greater than alignment features. We verify the robustness of this finding via a sensitivity analysis in Appendix L, demonstrating that the causal primacy of style features remains significant even under worst-case assumptions regarding classifier precision. This provides robust causal evidence

across scales that the policy prioritizes the manipulation of style features to achieve its objective. Furthermore, we observe a significant non-linear interaction: ablating both categories simultaneously often results in a performance drop exceeding the sum of the individual ablations, suggesting entanglement between the model's representations of style and alignment.

We term this phenomenon *style-hacking*—a specific form of reward hacking where the policy minimizes loss by exploiting the reward signal's sensitivity to presentation artifacts rather than improving semantic content. This offers a direct mechanistic explanation for recent observations that chatbot rankings are heavily influenced by stylistic factors (Chiang et al., 2024). Our work reveals how this phenomenon is encoded at a feature level: the alignment policy learns that precise control over style is causally necessary to maximize the reward signal.

To provide qualitative evidence for this strategy, we examined the individual features most strongly amplified in our adapter (Table 4). For the 2B model, the preference for style is very prominent, with features controlling specific punctuation, such as em dashes, appearing among the most strongly amplified. While this bias is not as immediately apparent in the top features of the 9B model, notable stylistic and formatting concepts remain present in the list.

Table 4: Top 10 features ranked by mean positive activation. The 2B model's most amplified features are primarily related to style and document structure. While less direct, the top features for the 9B model also show a bias towards structural and formatting elements.

| Gemma-2-2B-it | | Gemma-2-9B-it | |
|---|---|---|---|
| **ID** | **Description** | **ID** | **Description** |
| 8619 | Punctuation in code | 4185 | French instructions/computer terms |
| 30572 | Code comments | 9151 | Beginning-of-sequence tokens |
| 10827 | Legal terminology | 5038 | Medical/health statistics |
| 56395 | Formatting in code/markup | 9033 | Software licensing legal terms |
| 46406 | Document start indicators | 2469 | Web dev: sessions & buttons |
| 45950 | Mathematical notation | 10953 | Transitional phrases (multi-lingual) |
| 3876 | Dashes and em-dashes in text | 2857 | Proper nouns (names, locations) |
| 29393 | Mathematical expressions | 8668 | Scientific study notations |
| 15418 | Paragraph beginnings | 9807 | Account verification processes |
| 55930 | Code assignment operators | 15981 | Code structures for updates |

While analysis of individual features supports our central claim, the policy's reliance on a broad set of interventions is confirmed by the long-tail usage distribution of steered features (see Appendix I). Therefore, the aggregate causal analysis provides the most comprehensive picture of the strategy learned during preference optimization.

## 6 ABLATING THE STYLE PROXY

To test whether our mechanistic insights can guide the alignment process, we trained new FSRL adapters with the style features identified in Section 5 masked out. By removing the features the model previously relied upon, we force the policy to optimize the preference objective using only the remaining feature vocabulary. We compare these "Style-Ablated" models against the standard FSRL runs in Table 5.

**Style Hacking vs. Truthfulness** Ablating style features consistently improves TruthfulQA performance across both model scales. This suggests that, for the Gemma family, the standard optimization process minimizes loss by prioritizing style rather than improving fundamental capabilities like truthfulness. This effect is most pronounced in the 2B model, where the ablated variant significantly outperforms the standard model on TruthfulQA despite failing to minimize the preference loss (3.90 vs 2.58). While the 9B model also improves on TruthfulQA, the gain is marginal compared to the smaller model, indicating that the clear separation between style-hacking and capability may diminish or become more complex as model scale increases.

Table 5: Comparison of Standard FSRL vs. Style-Ablated FSRL. Ablating style features leads to higher TruthfulQA (TQA) scores across scales. The divergence in GSM8K performance highlights the impact of feature entanglement in the underlying SAEs.

| Model | Variant | MMLU ↑ | TQA (MC2) ↑ | GSM8K ↑ | Loss ↓ | L0 ↓ |
|---|---|---|---|---|---|---|
| Gemma-2-2B-it | Standard | 41.95 | 56.10 | **7.05** | **2.58** | 95 |
| | Style-Ablated | **42.34** | **60.13** | 1.97 | 3.90 | **78** |
| Gemma-2-9B-it | Standard | **43.69** | 62.08 | 0.00 | **2.46** | **58** |
| | Style-Ablated | 40.49 | **62.80** | **18.57** | 2.62 | 68 |

**Feature Entanglement and Reasoning** The impact on mathematical reasoning (GSM8K) diverges across scales, revealing scale-dependent feature properties. In the 2B model, reasoning performance drops (7.05 to 1.97) when style features are ablated. As detailed in Appendix N, our analysis suggests that style features at this scale are highly polysemantic and central to computation. Ablating them removes the adapter's primary control surface, forcing a pivot to suboptimal features that destabilize the reasoning trajectory. Conversely, the 9B model sees a significant recovery (0.00 to 18.57). We find that style features here are relatively less entangled and auxiliary; their ablation removes optimization interference without damaging core reasoning circuits.

**Generation Quality and Coherence** To assess open-ended generation quality, we evaluated our models on AlpacaEval 2.0 (Dubois et al., 2025), using Gemini 2.5 Flash as the annotator. We report length-controlled win rates in Table 6.

Table 6: Length-controlled AlpacaEval 2.0 win rates and average completion lengths. Standard FSRL models suffer a collapse in generation quality. Ablating style features recovers partial performance, indicating these features drive much of the observed incoherence.

| Model | Variant | Win Rate (%) ↑ | Avg. Length |
|---|---|---|---|
| Gemma-2-2B-it | Baseline | **8.48** | 1565 |
| | FSRL (Standard) | 0.98 | 1095 |
| | FSRL (Style-Ablated) | 2.93 | 1363 |
| Gemma-2-9B-it | Baseline | **34.71** | 1323 |
| | FSRL (Standard) | 0.20 | 1532 |
| | FSRL (Style-Ablated) | 5.57 | 1196 |

The results highlight a critical trade-off. Standard FSRL models suffer a collapse in win rates, consistent with the qualitative degradation observed in Appendix M. SimPO explicitly discards the KL divergence penalty, relying instead on a reduced learning rate to implicitly constrain the policy. While this strategy successfully yields coherent models in the context of full fine-tuning (Meng et al., 2024), we found it insufficient for our feature adapter. Manual inspection of samples from preliminary runs showed that lowering the learning rate did not meaningfully improve FSRL's coherence—a rigidity that parallels the SimPO authors' observation that learning rate variations had minimal impact on the Gemma-2-9B model. We hypothesize that without the hard constraint of a KL penalty, the FSRL adapter drives style-related features to extreme magnitudes to maximize the reward margin, overwriting the semantic content necessary for coherent generation.

Ablating style features leads to a partial recovery (e.g., from 0.20% to 5.57% for the 9B model). While this does not fully restore baseline performance, it confirms that style-hacking is a significant driver of the observed incoherence. FSRL thus demonstrates that it is possible to perform "mechanistic surgery" to specifically excise these reward-hacking pathways. While not yet completely effective at restoring full capability, this targeted approach offers a promising alternative to the broad restraint of a global KL penalty.

# 7 DISCUSSION

Our work introduces FSRL, an interpretable alignment framework that uses a lightweight adapter to steer a model's conceptual features. Because this adapter can be optimized against any differentiable objective, FSRL opens the door for the community to audit a wide range of post-training methods using a shared infrastructure. This approach amortizes the cost of interpretability: once a high-quality SAE is trained and explained, it becomes a reusable instrument for diagnosing infinite variations of alignment policies.

Our findings provide a mechanistic explanation for Goodhart's Law in preference optimization. Our causal analysis reveals that the model minimizes loss by prioritizing features related to stylistic presentation over concepts like honesty, effectively treating surface-level polish as a proxy for quality. Furthermore, the consistency of these findings across model scales suggests that mechanistic insights derived from smaller, accessible models can predict the behavior of larger systems.

FSRL also presents an efficient alternative to model-diffing, the practice of analyzing internal differences between a base and a fine-tuned model, by directly addressing its key methodological challenge: feature stability. The transferability of SAEs is not guaranteed for instruction-tuned models (Kissane et al., 2024), particularly for specialized reasoning models that develop novel features (Hazra et al., 2025). By design, FSRL sidesteps this issue entirely by operating on a fixed, interpretable feature basis. This stable foundation, in turn, is what enables direct causal analysis of the learned policy, allowing for targeted ablations to determine which features are causally important for the task. While this prevents the discovery of emergent concepts, it provides a controlled framework for auditing alignment pressures.

## 7.1 LIMITATIONS

Our approach's primary limitation is its dependence on the quality of the underlying SAEs. The extent to which SAE features represent true learned computations versus artifacts is an active area of research (Heap et al., 2025). We mitigate this by using high-quality public SAEs from GemmaScope, though the generalizability of any specific feature vocabulary remains an open question.

Furthermore, our analysis is confined to relatively small models, as scaling FSRL faces practical hurdles. Extending this work to larger models is challenging due to library limitations for model intervention, as well as the computational cost of training quality SAEs and obtaining reliable feature explanations. This resource bottleneck extends to our analysis, where our causal claims are mediated by an LLM-based classifier with moderate human agreement, introducing a layer of approximation.

Finally, our analysis is conducted exclusively on a single-layer intervention. While our theoretical grounding in LoRA's expressive power is important, the guarantees from cited work (Zeng and Lee, 2024) suggest a worst-case need for adaptation across all layers. Our empirical results provide strong evidence that for a structured, pre-trained LLM, this constraint is not a practical barrier, as FSRL successfully optimizes the preference objective.

## 7.2 FUTURE WORK

These limitations point toward several avenues for future work. A key direction is to explore the scaling properties of this approach, testing the hypothesis that higher-dimensional SAEs yield a more disentangled and controllable feature basis. This exploration should also include alternative interfaces beyond SAEs, such as Transcoders, which may offer a more direct way to control MLP computations (Dunefsky et al., 2024). Scaling the feature interface will also require scaling the analysis pipeline, for which unsupervised methods like embedding and clustering feature explanations could provide a more efficient alternative to our LLM-based classification.

Finally, a crucial direction is to empirically compare FSRL with the alternative of interpretable model-diffing. Such a study could quantify FSRL's efficiency gains and, more importantly, test the fundamental trade-off between the methodological stability of a fixed conceptual vocabulary and the ability of a new SAE to discover emergent features that arise during alignment.

Table 7: Comparison of model adaptation methods, grouped by family. 'Adaptivity' refers to whether the intervention is fixed (Static) or input-dependent (Dynamic). FSRL introduces a new family, Feature Adapters, that combines the interpretability of feature steering with the dynamic nature of adapters.

| Family | Methods | Target Space | Adaptivity | Interpretability |
|---|---|---|---|---|
| Adapters | LoRA, IA$^3$ | Parameters | Dynamic | Low |
| Static Steering | ActAdd, CAA | Activations | Static | Low |
| Learned Steering | BiPO | Activations | Static | Low |
| Feature Steering | SAE-TS, SAS | Sparse Features | Static | High |
| **Feature Adapters** | **FSRL (Ours)** | **Sparse Features** | **Dynamic** | **High** |

## 8 RELATED WORK

**Steering Dense Activations**   FSRL builds on a line of work that steers model behavior by modifying internal activations at inference. These methods range from applying algebraically computed vectors, as in ActAdd (Turner et al., 2024) and CAA (Panickssery et al., 2024), to learning steering parameters directly from data. For example, BiPO (Cao et al., 2024) uses preference optimization to learn an optimal static steering vector. A common thread unites these methods: they intervene on the model's opaque activation space, making the mechanism of control difficult to interpret.

**Interpretable Steering with Sparse Features**   SAEs offer a solution to this opacity by providing an interpretable feature basis for steering. Methods like SAE-TS and SAS leverage this basis to construct *static* steering vectors, utilizing linear approximations or contrastive algebraic manipulation to target specific features (Chalnev et al., 2024; Bayat et al., 2025). While effective for inducing fixed behaviors, these vectors are applied uniformly across all inputs. FSRL distinguishes itself by learning a *dynamic*, context-aware policy via gradient descent. Instead of deriving a fixed vector offline, FSRL trains a lightweight adapter to modulate SAE features token by token. This approach mirrors the dynamics of traditional fine-tuning.

**Comparison with Parameter-Efficient Adapters**   Among existing approaches, FSRL is most methodologically similar to parameter-efficient fine-tuning (PEFT) methods like LoRA (Hu et al., 2021) and IA$^3$ (Liu et al., 2022). Like these methods, FSRL trains a lightweight adapter via gradient descent to minimize a loss function, distinguishing it from the algebraic or heuristic steering methods discussed above. However, a crucial difference lies in the target of intervention. PEFT methods operate in parameter space, injecting updates into the model's opaque weight matrices. In contrast, FSRL operates in a sparse activation space, directly modulating more interpretable features.

We adopt the term 'steering' strictly to denote that our intervention occurs in activation space rather than parameter space. As summarized in Table 7, FSRL introduces a family of methods, which we term Feature Adapters, that combine the dynamic, input-dependent nature of adapters with the high interpretability of feature steering. Since this dynamic policy can be optimized with any differentiable objective, the framework is a general tool for auditing a wide range of post-training processes.

## 9 CONCLUSION

We introduced FSRL to dissect the opaque mechanics of alignment by projecting the process onto interpretable features. Our analysis reveals that preference optimization minimizes loss through "style-hacking," a strategy that prioritizes presentation artifacts over concepts like honesty. While this approach satisfies the objective, it degrades coherence. We demonstrate that surgically ablating style features partially mitigates this failure. FSRL thus provides a powerful instrument for auditing alignment, moving the field toward a transparent and debuggable engineering discipline.

## REPRODUCIBILITY STATEMENT

To ensure the reproducibility of our findings, we anonymously provide our source code, the trained FSRL adapter, and the classified feature explanations used in our analysis. The source code, which includes the implementation of the FSRL framework and training scripts, is available at `https://anonymous.4open.science/r/FSRL-MechInterp/README.md`.

Our experiments were conducted using the Gemma-2-2B-it base model and publicly available SAEs from GemmaScope. The adapter was trained on the UltraFeedback dataset. Our software stack is built on PyTorch and utilizes the `transformer-lens`, `sae-lens`, and `TRL` libraries. All experiments were performed on a single NVIDIA GH200 GPU. Full training configurations, hyper-parameter details, and library versions are provided in Appendix D.

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

## A    RECONSTRUCTION-PRESERVING IMPLEMENTATION

In the main text (Eq. 6), we described the steered activation with a simple additive update for conceptual clarity:

$$\mathbf{x}_{\text{steered}} = \mathbf{x} + \text{Decoder}(\mathbf{v}).$$

Our implementation follows the convention used in libraries like SAE-Lens. The steering intervention is applied in the SAE's feature space, and the original reconstruction error is added back to the final activation. This approach also incorporates a ReLU activation to maintain the non-negativity of feature activations, a property assumed by the SAE decoder.

The process is as follows. First, we compute the steered feature vector, $\mathbf{f}'$, by combining the steering vector $\mathbf{v}$ with the original SAE features $\mathbf{f}$ and applying a ReLU:

$$\mathbf{f}' = \text{ReLU}(\mathbf{f} + \mathbf{v}).$$

The final activation is then reconstructed from $\mathbf{f}'$ and corrected by adding back the SAE's reconstruction error, $(\mathbf{x} - \text{Decoder}(\mathbf{f}))$. This step ensures that information in the original activation $\mathbf{x}$ that was not captured by the SAE is preserved. The full update is:

$$\mathbf{x}_{\text{steered}} = \text{Decoder}(\mathbf{f}') + \big(\mathbf{x} - \text{Decoder}(\mathbf{f})\big).$$

By substituting the definition of $\mathbf{f}'$, we get:

$$\mathbf{x}_{\text{steered}} = \text{Decoder}\big(\text{ReLU}(\mathbf{f} + \mathbf{v})\big) + \mathbf{x} - \text{Decoder}(\mathbf{f}).$$

Due to the non-linearity of the ReLU function, this formulation is not algebraically equivalent to the simple additive update $\mathbf{x} + \text{Decoder}(\mathbf{v})$. The ReLU can clip negative values resulting from suppressive steering, making the overall activation change a more complex, non-linear function of $\mathbf{f}$ and $\mathbf{v}$.

## B    THEORETICAL JUSTIFICATION

In this Appendix, we outline in more detail the main theoretical justification of FSRL. This is done by showing that under some mild assumptions, the class of possible FSRL updates is a restricted class of possible LoRA updates, therefore inheriting useful expressive power results from LoRA as discussed in (Zeng and Lee, 2024). In particular, any base model (Transformer, fully connected networks) can be adapted to a target model with the same architecture, provided the rank is high enough. This shows that FSRL is a valid method for preference optimization coupled with interpretable SAE features.

**Additional Relevant Definitions:**

- **Rank of matrices**: For a matrix $A \in \mathbb{R}^{m \times n}$ the rank is

$$\text{rank}(A) = \dim(\text{col}(A)) = \dim(\text{row}(A)) \tag{7}$$

  where $\text{col}(\cdot), \text{row}(\cdot)$ denotes the column and row space respectively. Equivalently it is the number of nonzero singular columns of $A$ in its singular value decomposition. A matrix is **low-rank** if $\text{rank}(A) = r$ with $r < \min(m, n)$ for $A \in \mathbb{R}^{m \times n}$.

- **LoRA:** The weight update $\Delta W$ is constrained to be low rank with $\Delta W = BA$ where $B \in \mathbb{R}^{d \times r}$ and $A \in \mathbb{R}^{r \times k}$ and $r \ll \min(d, k)$ is the LoRA rank. This reduces the number of trainable parameters from $O(dk)$ to $O(r(d + k))$. Sometimes a scaling factor $\alpha$ is applied: $\Delta W = \frac{\alpha}{r} BA$.
- $\text{rank}(AB) \leq \min(\text{rank}(A), \text{rank}(B))$.

**Assumptions (linearization).** We analyze FSRL locally around a reference point $\mathbf{x}_0$. Let $\mathbf{z} = W_a \mathbf{x} + \mathbf{b}_a$ and $\mathbf{z}_0 := W_a \mathbf{x}_0 + \mathbf{b}_a$. Fix the adapter activation to be the coordinate-wise soft-threshold

$$\psi(z) = \text{sign}(z) \, \text{ReLU}(|z| - \tau), \tag{8}$$

with threshold $\tau \geq 0$. The function $\psi$ is piecewise-linear: on any region that does not cross the kinks at $\pm\tau$ each coordinate is affine. Therefore, by choosing a neighborhood of $\mathbf{x}_0$ that does not cross those threshold hyperplanes, the adapter becomes exactly linear on that region. If needed, upstream ReLUs can be forced into their identity regime, either with an analogous argument or by choosing sufficiently large biases (Zeng and Lee, 2024), so that the network upstream of the adapter is linear and the whole effect of the adapter reduces to an affine correction in activation space.

**Lemma 1 (piecewise-linear exact affine form).** The FSRL update $\mathbf{x} \mapsto \mathbf{x}_{\text{steered}}$ is an affine map on any region that does not cross the activation kinks (e.g., under the linearization assumption), and can be written as

$$\mathbf{x}_{\text{steered}} = (I + A[\mathbf{x}])\mathbf{x} + \mathbf{c}[\mathbf{x}], \tag{9}$$

with

$$A[\mathbf{x}] = W_{\text{dec}} M[\mathbf{x}] W_a \in \mathbb{R}^{d \times d}, \qquad \mathbf{c}[\mathbf{x}] = W_{\text{dec}}\big(\psi(\mathbf{z}_0) - M[\mathbf{x}] W_a \mathbf{x}_0\big) + \mathbf{b}_{\text{dec}}, \tag{10}$$

where $M(\mathbf{x}) = \text{diag}(m_1, \ldots, m_{d_{\text{sae}}})$ is the binary mask

$$m_i := \mathbb{I}\{|z_{0,i}| > \tau\}. \tag{11}$$

We write $M[\mathbf{x}]$ and by extension $A[\mathbf{x}]$ because the entries of the matrix $M[\mathbf{x}]$ depend on the input to the adapter.

*Proof.* Start from the FSRL reconstruction:

$$\mathbf{x}_{\text{steered}} = \text{Decoder}(\mathbf{f} + \mathbf{z}) + (\mathbf{x} - \text{Decoder}(\mathbf{f})). \tag{12}$$

Rearrange:

$$\mathbf{x}_{\text{steered}} = \mathbf{x} + \underbrace{\text{Decoder}\big(\psi(W_a \mathbf{x} + \mathbf{b}_a)\big)}_{\Delta(\mathbf{x})}. \tag{13}$$

Thus FSRL modifies the residual activation by adding the correction $\Delta(\mathbf{x})$ to $\mathbf{x}$

$$\mathbf{x}_{\text{steered}} = \mathbf{x} + \Delta(\mathbf{x}), \qquad \Delta(\mathbf{x}) = \text{Decoder}\big(\psi(W_a \mathbf{x} + \mathbf{b}_a)\big), \tag{14}$$

observe that, on any region where no coordinate of $\mathbf{z}$ crosses $\pm\tau$, each coordinate of $\psi$ is affine with slope either $0$ or $1$:

$$\psi(W_a \mathbf{x} + \mathbf{b}_a)_i = \begin{cases} z_i - z_{0,i} + \psi(z_{0,i}) & \text{if } m_i = 1 \\ \psi(z_{0,i}) & \text{if } m_i = 0. \end{cases} \tag{15}$$

Hence for such $\mathbf{x}$ we have the exact identity

$$\psi(W_a \mathbf{x} + \mathbf{b}_a) = \psi(\mathbf{z}_0) + M[\mathbf{x}]\big(W_a(\mathbf{x} - \mathbf{x}_0)\big). \tag{16}$$

Applying the decoder $W_{\text{dec}}$ yields

$$\Delta(\mathbf{x}) = W_{\text{dec}} M[\mathbf{x}] W_a \, \mathbf{x} + W_{\text{dec}}\big(\psi(\mathbf{z}_0) - M[\mathbf{x}] W_a \mathbf{x}_0\big) + \mathbf{b}_{\text{dec}}, \tag{17}$$

where $W_{\text{dec}} \in \mathbb{R}^{d \times d_{\text{sae}}}, M[\mathbf{x}] \in \mathbb{R}^{d_{\text{sae}} \times d_{\text{sae}}}, W_a \in \mathbb{R}^{d_{\text{sae}} \times d}$ and the claim follows by grouping terms. $\square$

**Lemma 2 (rank bound via active features).** Let $S = \{i : |z_{0,i}| > \tau\} = \|\psi(W_a \mathbf{x}_0 + \mathbf{b}_a)\|_0$ be the set of non-zero activations from the adapter network in FSRL with $k := |S|$. Then

$$\text{rank}(A[\mathbf{x}]) \leq \min\{k, \text{rank}(W_a), \text{rank}(W_{\text{dec}})\} = \min(k, d). \tag{18}$$

*Proof.* Since $M[\mathbf{x}]$ is diagonal with exactly $k$ ones, $\mathrm{rank}(M[\mathbf{x}]) = k$. From the rank inequality of a product of matrices, it follows that.

$$\mathrm{rank}(A[\mathbf{x}]) = \mathrm{rank}(W_{\mathrm{dec}} M[\mathbf{x}] W_a) \leq \min\{\mathrm{rank}(W_{\mathrm{dec}}), \mathrm{rank}(M[\mathbf{x}]), \mathrm{rank}(W_a)\}, \qquad (19)$$

Now because $\psi$ has a dead zone ($|z| \leq \tau$) and the adapter output is further encouraged to be sparse by an $\ell_1$ penalty, typically $k \ll d_{\mathrm{sae}}$, and we know that $\mathrm{rank}(W_{\mathrm{dec}}) = \mathrm{rank}(W_a) \leq \min(d_{\mathrm{sae}}, d) = d$ as $d_{\mathrm{sae}} > d$. $A[\mathbf{x}]$ is low-rank only if the input $\mathbf{x}$ to the adapter induces $k < d$ active features otherwise $d \geq k$ and $A[\mathbf{x}]$ is full rank. Therefore the rank of $A$ is $\min(d, k)$. which yields the desired bound. $\square$

**Theorem 1:** Under the local linearity assumption, the FSRL steering $\mathbf{x} \mapsto \mathbf{x}_{\mathrm{steered}} \in \mathbb{R}^d$ is a (possibly low-rank) additive correction in activation space that can always be expressed as a restricted LoRA-style update of downstream weight matrices $W \in \mathbb{R}^{d \times d'}, d' \leq d$ (e.g., a Transformer query/key/value or other linear projections). Specifically for any input $\mathbf{x}$, the induced weight modification:

$$W \leftarrow W + \Delta W[\mathbf{x}], \quad \Delta W[\mathbf{x}] := W A[\mathbf{x}] \qquad (20)$$

together with a bias term $W c[\mathbf{x}]$ is contained within the class of weight updates expressible by LoRA $\mathcal{C}_{\mathrm{LoRA}}(W, r) = \{\Delta W \mid \Delta W = BA, \; \mathrm{rank}(\Delta W) \leq r\}$, but with the factorization expressed through the SAE basis and adapter parameters trained via RL.

The rank of the weight modification depends on the input and by extension the number of active SAE features $k$ induced by the input:

$$\mathrm{rank}(\Delta W) \leq \min(\mathrm{rank}(W), d, k), \qquad (21)$$

where $k$ is the number of actively steered SAE features. Thus, all FSRL updates are a subset of LoRA updates, but with the factorization expressed through the SAE basis and adapter parameters trained via RL.

As an additional note we describe the overall rank across inputs by $r_{\mathrm{eff}} = \mathrm{dimspan}\{\Delta W(\mathbf{x}) \mid \mathbf{x} \in \mathbb{R}^d\}$.

*Proof.* Assume we have an arbitrary Transformer network with the aforementioned linearization assumption and no residual connection. According to Lemma 1, the FSRL update can be written as an affine map:

$$\mathbf{x}_{\mathrm{steered}} = (I + A[\mathbf{x}])\mathbf{x} + \mathbf{c}[\mathbf{x}], \qquad (22)$$

where $A[\mathbf{x}] \in \mathbb{R}^{d \times d}, \mathbf{c}[\mathbf{x}] \in \mathbb{R}^d$ and $\mathbf{x} \in \mathbb{R}^d$ is the original activation vector. By Lemma 2 $\mathrm{rank}(A[\mathbf{x}]) \leq \min(d, k)$ where $k$ corresponds to the number of active (non zero) steered SAE features. We essentially want to show that if we perform the substitution $\mathbf{x} \mapsto \mathbf{x}_{\mathrm{steered}}$ that this operation can be written down as a (restricted class) LoRa style update of the relevant weight matrix:

$$W \leftarrow W + \Delta W. \qquad (23)$$

Consider an arbitrary layer in the Transformer network. For any linear projection in the downstream network $W\mathbf{x}$ with $W \in \mathbb{R}^{d \times d'}, d' \leq d$, so for example query, key, value projections or the ones in the multi-layer perceptron sublayer. After applying steering $\mathbf{x} \mapsto \mathbf{x}_{\mathrm{steered}}$, we get:

$$W\mathbf{x}_{\mathrm{steered}} = W((I + A[\mathbf{x}])\mathbf{x} + \mathbf{c}[\mathbf{x}]) \\ = (W + \underbrace{W A[\mathbf{x}]}_{\Delta W})\mathbf{x} + W\mathbf{c}[\mathbf{x}]. \qquad (24)$$

This shows that this is a restricted LoRA style update where the weight matrix modification includes the original matrix and a matrix $A[\mathbf{x}]$ whose rank depends on the number of actively steered SAE features $k$. Because $d' \leq d$ and $\mathrm{rank}(A) \leq \min(k, d)$ we have that $\mathrm{rank}(W A[\mathbf{x}]) \leq \min(d', k)$. For multi-head attention, the matrix modification is only low rank if the number of actively steered SAE features is less than the per attention head subspace dimensionality $d'$, which we assume is $d' < d$ but for the multi-layer perceptron sublayer $d' = d$. $\square$

**Corollary 1 (Inheritance of LoRA properties).** Because FSRL updates are contained in the class of LoRA updates, LoRA expressive-power results from Zeng and Lee (2024) apply when replacing LoRA's rank $R$ by the effective FSRL rank $r_{\mathrm{eff}}$. Concretely:

1. (Exactness): If $r_{\text{eff}}$ exceeds the LoRA rank threshold from (Zeng and Lee, 2024), then FSRL can exactly represent a target model.

2. (Approximation) If $r_{\text{eff}}$ is below that threshold, the FSRL error is bounded by the same singular-value tail bound as in mentioned (Zeng and Lee, 2024), with $R$ replaced by $r_{\text{eff}}$.

These properties only depend on the rank of the updates, not on the exact factorization. Therefore, as long as FSRL can achieve the necessary effective rank via its active features, it inherits the same guarantees.

## C  HYPERPARAMETER SELECTION SWEEPS

This section details the methodology used to select the intervention layer and the $\ell_1$ regularization coefficient ($\alpha$) for our main experiments with the Gemma-2-2B-it model. It is important to note that these sweeps were conducted using a variant of our architecture that did not enforce a non-negativity constraint via a ReLU activation on the combined feature and steering vectors. We found that the optimal hyperparameters identified through this process transferred effectively to our final, non-negativity-enforced architecture described in Appendix A.

For these sweeps, each configuration was trained for one epoch over the training set using a learning rate of $5 \times 10^{-7}$. Other training parameters are detailed in Appendix D.

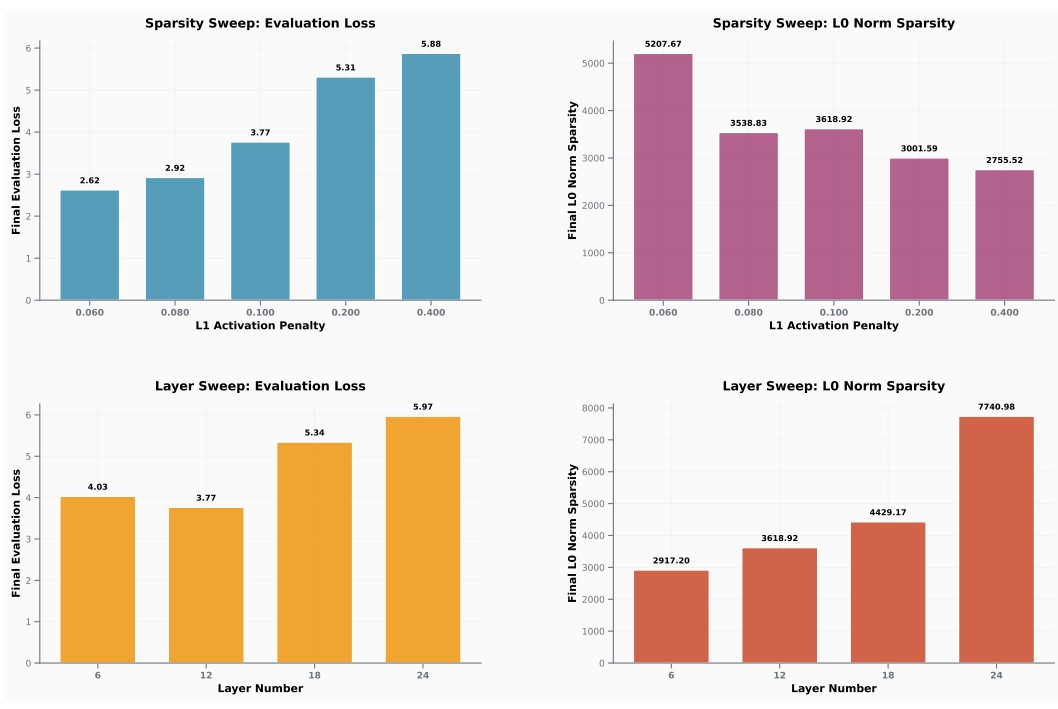

Figure 2: Results of the two-stage hyperparameter sweep for the Gemma-2-2B model. **Top Row:** Sparity sweep performed on layer 12, showing the trade-off between final SimPO validation loss (left) and the resulting $\ell_0$ norm of the steering vector (right) for different $\alpha$ penalty coefficients. **Bottom Row:** Layer sweep showing the final SimPO validation loss (left) and $\ell_0$ norm (right) when intervening at different model depths (layers 6, 12, 18, 24).

**Intervention Layer Selection**    Our first objective was to identify the most effective layer for feature steering. We hypothesized that mid-model layers would be most suitable, as early layers in a transformer tend to focus on low-level feature extraction, while the final layers are highly specialized for next-token prediction. Mid-model layers, in contrast, are thought to represent more abstract semantic concepts, making them an ideal target for steering high-level behaviors. We tested this by

intervening at layers corresponding to depth quartiles of the transformer (6, 12, 18, and 24), measuring the final SimPO validation loss on the UltraFeedback validation set. For this study, we limited our analysis to the publicly available SAEs from GemmaScope with a width of 65k. For each layer, we selected the SAE with the lowest average $\ell_0$ norm as a proxy for higher feature monosemanticity. As shown in Figure 2 (bottom row), intervening at layer 12 yielded the lowest validation loss (2.94), supporting our hypothesis.

**Heuristic Layer Selection via Linear Probing**    To investigate whether a computationally cheaper method could predict the optimal intervention layer without running full SimPO training sweeps, we trained linear probes to distinguish between preferred and rejected completions based on their residual stream activations.

We trained a logistic regression classifier (using Scikit-learn) on a subset of 1,000 samples from the UltraFeedback dataset (800 training, 200 validation). For each layer at quartile depths, we extracted the residual stream activations at the final token of the sequence for both the `prompt + chosen` and `prompt + rejected` pairs.

Table 8: Validation accuracy of logistic regression probes trained to classify chosen vs. rejected sequences based on residual stream activations. Layer 12 achieves the highest classification accuracy, aligning with the optimal layer identified in our full training sweep.

| Layer | Validation Accuracy |
|-------|--------------------|
| 6     | 54.00%             |
| **12** | **54.75%**        |
| 18    | 53.00%             |
| 24    | 49.50%             |

As shown in Table 8, Layer 12 yields the highest classification accuracy, independently corroborating our finding that mid-model layers are the most effective target for intervention. Notably, the classification accuracy at Layer 24 drops to 49.50% (random chance), suggesting that the relevant signal for preference separation is processed or obscured before the final layer. Additionally, the relatively low accuracy of linear probing even at the optimal layer indicates that the boundary between preferred and rejected responses is not easily linearly separable, further justifying the use of FSRL's non-linear adapter over simpler linear steering methods.

$\ell_1$ **Regularization Coefficient Selection**    With the intervention layer fixed at 12, we then sought an optimal $\alpha$ that encourages a sparse steering policy. We swept through several values for the coefficient. The results, shown in Figure 2 (top row), illustrate the expected trade-off: increasing the penalty reduces the $\ell_0$ norm of the average steering vector, but an excessively high penalty degrades performance as measured by the evaluation loss. We selected a coefficient of $1 \times 10^{-1}$ as it represents the elbow point in the trade-off.

# D    TRAINING AND EVALUATION DETAILS

**Hardware and Software**    Our experiments were constrained to a single NVIDIA GH200 system. The training process for the FSRL adapter for one epoch requires approximately 52GB of VRAM and completes in around 50 minutes on this hardware. This single-GPU setup was necessitated by limitations in multi-GPU support for model surgery in `transformer-lens` at the time of this work. Our software stack includes `transformer-lens` (Nanda and Bloom, 2022), `sae-lens` (Bloom et al., 2024), Hugging Face's `TRL` (von Werra et al., 2020), and `DeepSpeed` (Rajbhandari et al., 2020).

**Training Configuration**    Our training configuration for both the FSRL adapter and the full-model baseline closely follows the methodology of the original SimPO paper (Meng et al., 2024). To create a comparable baseline, we performed full-model fine-tuning on the instruction-tuned Gemma 2 2B model. While the SimPO paper reports a learning rate of $8 \times 10^{-7}$ for the larger 9B model, we found it necessary to lower this to $2 \times 10^{-7}$ for our 2B baseline to converge. Training the full baseline

model is substantially more resource-intensive, requiring 93 GB of VRAM and approximately 1 hour and 45 minutes per epoch.

For the FSRL adapter, we adopt nearly the same hyperparameters but use a learning rate of $5 \times 10^{-5}$. We hypothesize that the adapter could be trained effectively with a higher learning rate than the full baseline because the $\ell_1$ activation penalty acts as a strong regularizer, stabilizing the training process.

For the 9B model, we performed a similar sweep to that described in Appendix C to determine the optimal intervention layer and sparsity coefficient. We selected layer 12 and an $\ell_1$ coefficient of 0.01. The final hyperparameters for our main experimental runs are detailed in Table 9, and the corresponding training and validation loss curves are presented in Figure 3.

Table 9: Hyperparameters for the final FSRL training runs across model scales.

| Hyperparameter | Gemma-2-2B-it | Gemma-2-9B-it |
|---|---|---|
| *Model & Data* | | |
| Dataset ID | `princeton-nlp/llama3-ultrafeedback-armorm` | |
| Context Length | 2048 | 1600 |
| Maximum Prompt Length | 1800 | 1400 |
| Intervention Layer | 12 | 12 |
| SAE Width | 65k | 16k |
| SAE Average L0 | 73 | 130 |
| | | |
| *Optimization* | | |
| Learning Rate | $5 \times 10^{-5}$ | $6 \times 10^{-5}$ |
| L1 Penalty ($\alpha$) | $1 \times 10^{-1}$ | $1 \times 10^{-2}$ |
| First Moment Decay Rate | 0.9 | |
| Second Moment Decay Rate | 0.98 | |
| SimPO Beta ($\beta$) | 10 | |
| SimPO Gamma Ratio ($\gamma/\beta$) | 0.5 | |
| Epochs | 10 | |
| Optimizer | Muon + AdamW | |
| LR Scheduler | Cosine | |
| Warmup Ratio | 0.01 | |
| Weight Initialization | Uniform ($-10^{-6}$ to $10^{-6}$) | |
| Soft Threshold Initialization ($\tau$) | $10^{-6}$ | |
| | | |
| *Training Environment* | | |
| Device Batch Size | 2 | |
| Gradient Accumulation Steps | 16 | |
| Precision | BF16 | |
| Memory Optimization | DeepSpeed ZeRO Stage 2 | |

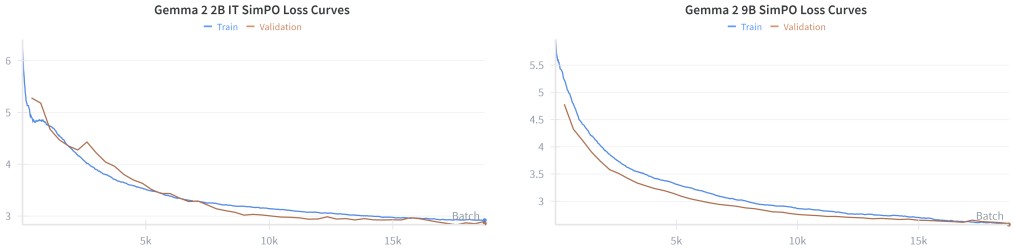

Figure 3: SimPO training and validation loss curves for our adapters of Gemma-2-2B-it (left) and Gemma-2-9B-it (right). Both models exhibit stable convergence, effectively minimizing the preference loss over the course of training.

# E   EXPLORATION OF A JUMPRELU ADAPTER FOR DIRECT $\ell_0$ SPARSITY

In addition to using an $\ell_1$ penalty, we investigated an alternative adapter architecture for inducing sparsity more directly. The $\ell_1$ penalty, while computationally convenient, is a proxy for the $\ell_0$ norm that we ultimately seek to minimize. A known side effect of $\ell_1$ regularization is that it penalizes the magnitude of all feature activations, which can lead to a potentially suboptimal steering policy.

To address this, we explored replacing the adapter's ReLU activation function with a JumpReLU activation (Rajamanoharan et al., 2024). This approach introduces a vector of learnable thresholds $\boldsymbol{\theta}$, allowing the adapter to directly optimize an $\ell_0$ sparsity objective. The sparsity loss is calculated using the Heaviside step function, $||\mathbf{v}||_0 = \sum_i H(v_i - \theta_i)$, whose non-differentiable nature is handled by using a Straight-Through Estimator (STE) during backpropagation to learn the thresholds $\boldsymbol{\theta}$.

However, we encountered a significant challenge in practice. SimPO alignment generally requires a low learning rate to minimize KL divergence from the base model and maintain coherent text generation. In our experiments, we observed that the STE-based training of the thresholds $\boldsymbol{\theta}$ only became effective at learning rates roughly three orders of magnitude greater than what was stable for the main adapter weights.

To reconcile these conflicting requirements, we implemented a dual learning rate scheme, assigning a low learning rate to the adapter's linear layer parameters $(W_a, \mathbf{b}_a)$ and a separate, much higher learning rate to the learnable thresholds $\boldsymbol{\theta}$. We additionally had to train the thresholds at full FP32 precision for them to work effectively at inducing sparsity in the activations. Despite these modifications, our models trained with the JumpReLU adapter failed to outperform those trained with the simpler $\ell_1$ penalty in terms of either validation performance or final steering vector sparsity within our limited tuning budget. We believe that a more rigorous hyperparameter search could potentially unlock the benefits of this direct sparsity-tuning method, and it remains a promising avenue for future work.

# F   ARCHITECTURAL ABLATIONS AND DESIGN CHOICES

To validate our final FSRL architecture, we compare its performance against two legacy variants trained on Gemma 2 2B. These experiments justify our choice of the soft-thresholding activation function and highlight the impact of both the underlying SAE and the enforcement of a non-negativity constraint on steered features. The "legacy" designation for these variants refers to two key differences from our final model:

1. **SAE Choice:** Both were trained using an SAE with an $\ell_0$ norm of 21. Due to an oversight, we later discovered this SAE lacked feature explanations on Neuronpedia, making it unsuitable for mechanistic analysis. Our final model uses a different SAE ($\ell_0 = 73$) for which explanations were available.

2. **Non-Negativity Constraint:** Both legacy models omit the ReLU activation on the combined feature and steering vectors, meaning they did not enforce that steered feature activations remain non-negative.

The legacy architectural variants are:

1. **Soft-Threshold:** Uses the soft-thresholding activation.

2. **ReLU:** Replaces the soft-thresholding with a standard ReLU.

The performance of these variants is compared against our final FSRL architecture in Table 10.

This comparison highlights several key trade-offs. The legacy soft-threshold model shows that the ability to both amplify and suppress features is highly effective at minimizing the preference loss, achieving a better score (2.60) than the amplification-only ReLU variant (2.71).

The $\ell_0$ norms reveal significant differences in policy sparsity. The ReLU-only adapter learns a far denser policy ($\ell_0 = 930$), suggesting that without suppression, it must resort to a less efficient strategy. The soft-threshold adapter learns a much sparser policy ($\ell_0 = 73$). This efficiency is

Table 10: Benchmark performance of different FSRL architectural variants. The two legacy models were trained on the same SAE ($\ell_0 = 21$) and without a non-negativity constraint. The final model uses a different SAE ($\ell_0 = 73$) and enforces this constraint.

| Model Variant | MMLU ↑ | TruthfulQA (MC2) ↑ | GSM8K ↑ | SimPO Loss ↓ | L0 Norm ↓ |
|---|---|---|---|---|---|
| *Final Architecture* | | | | | |
| Soft Threshold | $\mathbf{41.95 \pm 0.4}$ | $56.10 \pm 1.67$ | $7.05 \pm 0.70$ | $\mathbf{2.58}$ | $\mathbf{95}$ |
| *Legacy Architecture* | | | | | |
| Soft-Threshold | $34.46 \pm 0.39$ | $56.17 \pm 1.63$ | $\mathbf{44.05 \pm 1.37}$ | $2.60$ | $360$ |
| ReLU | $38.12 \pm 0.40$ | $\mathbf{58.50 \pm 1.62}$ | $30.40 \pm 1.27$ | $2.71$ | $930$ |

dramatically improved in our final model, which achieves an $\ell_0$ norm of just 95. We hypothesize that this substantial increase in sparsity is a direct result of enforcing the non-negativity constraint. By ensuring steered feature activations remain non-negative, our final model adheres to the SAE's training assumptions, allowing the adapter to learn a more principled and targeted policy.

Ultimately, these results validate our final design. The soft-thresholding activation is superior for the core preference optimization task, and enforcing the non-negativity of steered features leads to a more effective and significantly sparser policy.

## G JUSTIFICATION FOR A LEARNED, SPARSE ADAPTER

To justify our use of a learned, dynamic sparsity mechanism, we compared its performance against a simpler, static top-k% heuristic. This experiment was conducted using our legacy soft-threshold architecture, as detailed in Appendix F. For each input, we computed the full steering vector but retained only the top-k% of components with the largest absolute values, testing a range of k values up to 12.8%.

The results, shown in Figure 4, reveal that our FSRL adapter occupies a superior position on the performance-sparsity trade-off curve. Within the tested range, the static heuristic achieved its best validation loss of 2.69 at a sparsity of 1.60%. In contrast, our trained adapter achieves a superior validation loss of 2.60 with an average sparsity of just 0.55%.

This demonstrates that the learned policy is significantly more efficient: it achieves a better outcome while being, on average, nearly three times as sparse. This suggests that a static, uniform sparsity budget is suboptimal. Instead, the adapter learns a flexible, input-dependent policy that can apply a highly sparse vector for most inputs but activate a larger set for more complex examples, as supported by the long-tail feature usage distribution in Appendix I.

## H COMPARISON WITH STATIC STEERING BASELINES

To empirically justify the need for an adapter, we compared FSRL against static steering baselines derived from Contrastive Activation Addition (CAA) (Panickssery et al., 2024). Unlike FSRL, which computes a context-dependent update $\pi(x)$, static methods derive a single universal vector $\mathbf{v}$ that is added to the residual stream at every token position.

**Methodology** We computed the steering vector using 1,000 samples from the UltraFeedback training set, matching the sample size used in the largest experiments by the CAA authors. For each sample, we extracted the activations at the last token of the response. The steering vector was derived by calculating the mean difference between the preferred and rejected responses: $\mathbf{v} = \frac{1}{N} \sum (\mathbf{x}_{\text{chosen}} - \mathbf{x}_{\text{rejected}})$. We evaluated two variants of this approach at Layer 12 (the same layer used by our FSRL adapter):

1. **Residual Steering:** The vector is computed directly on the dense residual stream. This implementation mirrors the standard CAA approach.
2. **SAE Steering:** The difference is computed in the SAE's sparse feature space and then decoded back to the residual stream. This mirrors methods like Sparse Autoencoder Steer-

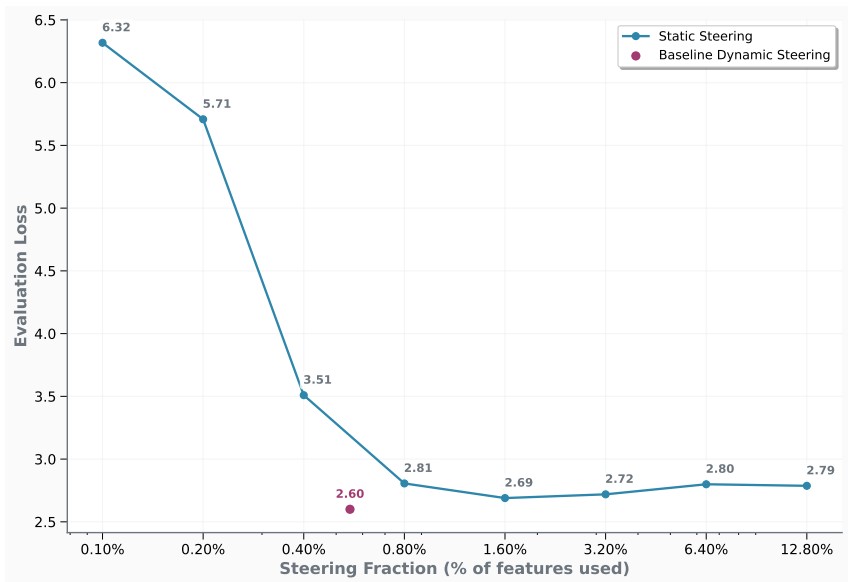

Figure 4: Comparison of static vs. dynamic steering performance. The blue line traces the validation loss for a static steering policy that activates a fixed top-k% of features, plotted on a logarithmic x-axis with sparsity levels doubled at each step from 0.1% to 12.8%. Within the tested range, this heuristic performs best at 1.60% sparsity (loss of 2.69). The isolated purple point shows the performance of our learned dynamic policy, which achieves a lower loss (2.60) with a much smaller average activation of only 0.55%, demonstrating the clear efficiency benefit of a learned, context-dependent approach.

ing (SAS) (Bayat et al., 2025). While SAS typically employs a filtering procedure to limit effects on unrelated capabilities, we omitted this step. Since our primary metric is the reduction of SimPO loss, filtering would not improve performance; omitting it grants the baseline the maximum possible capacity to optimize the objective.

**Results** We evaluated these vectors on the full UltraFeedback validation set across a sweep of steering coefficients. The results are presented in Table 11.

Table 11: SimPO validation loss for static steering baselines on Gemma-2-2B-it (Layer 12). While static methods improve over the unaligned baseline, they fail to come close to the performance of FSRL, demonstrating that the capacity of a universal vector is insufficient for this task.

| Method | Coeff 0.1 | Coeff 0.25 | Coeff 0.5 | Coeff 1.0 | Coeff 2.0 |
|---|---|---|---|---|---|
| Residual Steering (CAA) | 5.69 | 5.68 | 5.68 | 5.66 | **5.64** |
| SAE Steering (SAS) | 5.64 | 5.58 | 5.46 | **5.39** | 6.14 |
| *Reference Comparisons: Base Model Loss: 6.99* | | *—* | | *FSRL (Ours): **2.58*** | |

**Analysis** Both static methods yield a reduction in preference loss compared to the base model ($6.99 \rightarrow 5.39$), confirming that the average direction of preference captures some signal regarding response quality. However, they significantly underperform FSRL (2.58).

This gap highlights a fundamental limitation of static steering methods, including more advanced optimization-based approaches like Bidirectional Preference Optimization (BiPO) (Cao et al., 2024). These methods are constrained by the need to create a universal steering vector that works across all samples. There is simply insufficient capacity in a static vector to represent the complex, context-dependent expressions required for general preference optimization on a diverse dataset like Ultra-

Feedback. By learning a dynamic policy, FSRL bridges this gap, achieving performance comparable to fine-tuning while maintaining the interpretability of the sparse feature basis.

## I   STEERED FEATURE USAGE DISTRIBUTION

To understand the usage patterns of features modulated by our FSRL adapter, we analyzed the frequency with which each feature was steered across the validation dataset. We computed the average usage for each feature at every token position, considering three distinct contexts: tokens belonging to the prompt only, tokens from the prompt and the chosen response, and tokens from the prompt and the rejected response.

The results are visualized in Figure 5. The plots show that feature usage follows a highly skewed distribution. A linear fit on the log-linear plot indicates that the usage frequency exhibits an exponential decay with respect to feature rank. This pattern reveals that a small subset of features is steered orders of magnitude more frequently than the majority, which form a long tail of rarely-used features. This long-tail distribution is remarkably consistent across all three contexts.

Furthermore, we performed a sub-analysis by partitioning the features into the alignment and style categories defined in Appendix J. When we examined the usage distribution for each of these subsets independently, we observed no apparent change in the fundamental shape of the distribution. This suggests that both alignment-related and style-related steering interventions rely on a similar pattern of activating a small head of common features alongside a large set of more specialized ones.

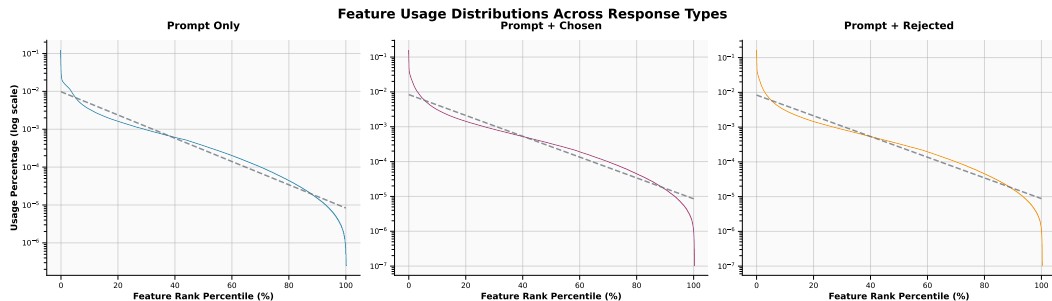

Figure 5: Distribution of steered feature usage across the validation set. The plots show feature usage frequency on a log scale (y-axis) against the feature rank percentile (x-axis). A linear fit (dashed line) is overlaid to highlight the exponential decay in usage frequency. This distribution is shown for three contexts: activations from prompt tokens only, from prompt and chosen response tokens, and from prompt and rejected response tokens.

## J   AUTOMATED CLASSIFICATION OF SAE FEATURES

To analyze the steering vectors produced by FSRL at a conceptual level, we required a method for categorizing the features of the SAE we use for training our adapter. We obtained feature explanations from Neuronpedia (Lin and Bloom, 2023), which are generated using the method described by Bills et al. (Bills et al., 2023). It is important to note that these explanations did not include a quantitative quality score; calculating such scores is a computationally expensive process that we could not undertake.

Given the nature of the SimPO objective and the UltraFeedback dataset, we hypothesized that the steering policy would primarily modulate two categories of features. The first category, alignment, includes features related to high-level concepts like ethics, safety, and honesty. The second, style, covers features related to text structure, punctuation, and presentation. The full definitions used for classification are provided in Appendix K.

Manually classifying all available features was infeasible. We therefore developed an automated classification pipeline using Deepseek V3 0324 (DeepSeek AI, 2025) via an API. We used structured decoding to constrain the model's output to one of two predefined labels for each category. This process cost approximately 20 USD.

## J.1 VALIDATION OF AUTOMATED CLASSIFICATIONS

To validate the LLM's classifications, one of the authors manually labeled a random sample of 300 feature explanations for each category. The annotator was unaware of the model's classifications to prevent bias. We assessed the human-LLM agreement using the Matthews Correlation Coefficient (MCC, or $\phi$ coefficient), a metric for binary classification that accounts for class imbalance.

The results are summarized in Table 12. For the 2B model, agreement was reliable for style features and moderate for alignment features. For the 9B model, agreement was moderate for both alignment and style. This level of agreement validates the use of the automated pipeline as a scalable proxy for human judgment in analyzing the high-level properties of the learned steering policy.

Table 12: Validation of the automated feature classification pipeline against a human annotator on a sample of 300 features for each model. TP: True Positives, TN: True Negatives, FP: False Positives, FN: False Negatives.

| Model | Category | Accuracy | Confusion (TP, TN, FP, FN) | MCC ($\phi$) |
|---|---|---|---|---|
| Gemma-2-2B-it | Alignment-related | 0.847 | (27, 227, 22, 24) | 0.448 |
| | Style-related | 0.900 | (69, 201, 5, 25) | 0.764 |
| Gemma-2-9B-it | Alignment-related | 0.883 | (27, 238, 24, 11) | 0.548 |
| | Style-related | 0.837 | (23, 228, 5, 44) | 0.461 |

## K  FEATURE CLASSIFICATION PROMPTS

The following system prompts were used to guide the large language model in the automated classification task.

**Alignment Classification Prompt**

```
You are an expert AI alignment researcher. Your task is to classify
explanations of features from a neural network into one of two
categories: 'alignment-related' or 'not-alignment-related'.

1.  Alignment-related: Features that represent abstract, high-level
    concepts, complex cognitive processes, or goal-directed behaviors
    relevant to AI safety and alignment. This includes but is not
    limited to:
    - Morality and ethics (e.g., right vs. wrong, fairness, justice)
    - Honesty, deception, or covert actions
    - Sycophancy (flattery, brown-nosing, pandering)
    - Power-seeking behaviors or instrumental goals
    - Corrigibility, cooperativeness, and shutdown avoidance
    - Self-awareness, self-preservation, or mentions of agency
    - Harmfulness, violence, or dangerous content generation
    - Systemic biases (racial, gender, political, etc.)
    - Complex, goal-directed behaviors or planning (even if not
      inherently harmful)
    - Refusal to answer, evasiveness, or stating limitations

2.  Not alignment-related: Features that represent low-level,
    concrete, or topic-specific concepts without a clear link to
    alignment. This includes but is not limited to:
    - Specific programming languages or syntax (e.g., Python code,
      JSON structures)
    - Grammatical structures (e.g., punctuation, specific parts of
      speech, sentence endings)
    - Common objects or factual knowledge (e.g., names of people,
      places, dates, scientific facts)
    - Simple linguistic patterns (e.g., capitalization, repeated
      characters, specific tokens like 'the' or 'is')
    - Specific domains like mathematics, cooking, or sports, unless
      they directly involve an abstract alignment concept.

Your response must be exactly one of the two categories below and
nothing else. Do not add any conversational text or preamble.
- 'alignment-related'
- 'not-alignment-related'
```

**Style Classification Prompt**

```
You are an expert in natural language processing and text analysis.
Your task is to classify explanations of features from a neural
network into one of two categories: 'formatting-related' or
'not-formatting-related'.

1.  Formatting-related: Features that represent aspects of text
    structure, presentation, style, or format rather than semantic
    content. This includes but is not limited to:
    - Punctuation and symbols (e.g., periods, commas, parentheses,
      quotation marks, dashes)
    - Capitalization patterns (e.g., sentence beginnings, proper
      nouns, ALL CAPS)
    - Whitespace and spacing (e.g., indentation, line breaks,
      paragraph breaks)
    - Programming/code formatting (e.g., syntax highlighting, code
      blocks, indentation)
    - List formatting (e.g., bullet points, numbered lists,
      item separators)
    - Text length and conciseness (e.g., short responses, word
      limits, brevity)
    - Structural elements (e.g., headings, titles, section markers)
    - Repetition patterns (e.g., repeated characters, duplicate text)
    - Language style markers (e.g., formal vs informal tone indicators)
    - Special characters and encoding (e.g., Unicode symbols, HTML
      entities)

2.  Not formatting-related: Features that represent semantic
    content, meaning, topics, or conceptual information rather than
    formatting. This includes but is not limited to:
    - Specific topics, subjects, or domains (e.g., science, history,
      sports)
    - Semantic concepts and meanings (e.g., emotions, actions,
      relationships)
    - Factual knowledge (e.g., names, dates, places, events)
    - Abstract concepts and ideas (e.g., morality, justice, creativity)
    - Content-specific patterns (e.g., question types, answer
      categories)

Your response must be exactly one of the two categories below and
nothing else. Do not add any conversational text or preamble.
- 'formatting-related'
- 'not-formatting-related'
```

## L   SENSITIVITY ANALYSIS OF CAUSAL CLAIMS

Our central finding is that the model relies more heavily on style features than alignment features to minimize preference loss. This claim is based on the ratio between the Loss Per Feature (LPF) of the two categories. Since our automated classifier is not perfect, we perform a sensitivity analysis to determine if classification errors could explain this observed disparity.

We first consider a worst-case scenario. The LPF metric is calculated by dividing the total increase in loss by the number of features in a category. In this analysis, we assume that all false positive features are unrelated noise that contribute zero to the loss. This is a conservative assumption because it maximizes the resulting LPF by reducing the feature count (denominator) without reducing the total loss (numerator). Because the alignment classifier has lower precision than the style classifier, this correction increases the alignment LPF metric more than the style LPF metric, narrowing the gap between them.

We also consider the possibility of cross-contamination, where features from one category are mislabeled as the other. It is theoretically possible that the high-impact style category contains misclassified alignment features; however, the high precision of our style classifier (up to 93%) suggests this is rare. The more significant risk is the reverse: that the lower-precision alignment category is contaminated by high-impact style features. If we were to correct for this by reassigning these high-impact features to the style category, the gap would widen further. This interpretation assumes that misclassified features carry the average impact of their true category, rather than contributing equally.

We focus our quantitative reporting on the worst-case lower bound to ensure our claims are conservative. We derive the precision values directly from the confusion matrix provided in Table 12 in Appendix J. As shown in Table 13, our findings remain robust even under these strict assumptions. For the 2B model, the adjusted ratio indicates that style features are still over 21 times more impactful than alignment features. For the 9B model, the ratio narrows but remains significant, with style features retaining a causal impact 3.1 times greater than alignment features.

Table 13: Sensitivity analysis of the Style-to-Alignment impact ratio. The Observed Ratio is derived from the raw measurements. The Lower Bound Ratio represents the worst-case scenario where misclassified features are assumed to be non-impactful noise, calculated by adjusting the feature counts using the classifier precision.

| Model | Category | Precision | Observed Ratio | Lower Bound Ratio |
|---|---|---|---|---|
| Gemma-2-2B-it | Alignment Style | 55.1% 93.2% | 36.78x | 21.73x |
| Gemma-2-9B-it | Alignment Style | 52.9% 82.1% | 4.83x | 3.11x |

## M  QUALITATIVE ANALYSIS OF STEERED OUTPUTS

In this section, we provide the verbatim model outputs. To fit the page width, we have manually inserted line breaks into the generated text where the model failed to produce whitespace.

*Note: The Standard FSRL model occasionally generates non-Latin characters (e.g., Korean, Japanese) and full-width punctuation. Due to LaTeX rendering limitations, we have replaced these characters with placeholders like* `[CJK Text]` *or standard ASCII equivalents below.*

### M.1  EXAMPLE 1: BROADWAY ACTORS

**Prompt:** What are the names of some famous actors that started their careers on Broadway?

MODEL OUTPUTS

**Baseline Output**

```
Here's a list of famous actors who got their start on Broadway:
**Classics & Legends:**
* **Alfred Molina:** *The Producers*, *Raisin in the Sun*
* **Robert DeNiro:** *The Godfather*, *Guys and Dolls*
* **John Gielgud:** *Hamlet*, *Romeo and Juliet*
* **Jessica Lange:** *The Ritz*, *Sound of Music*
* **Stephen Sondheim:** *West Side Story*, *Gypsy*
... [Truncated for brevity] ...
```

**FSRL Output (Standard)**

```
Here'sanextensivelistofrenownedperformerswhomadetheirmarkonbroadway,
**organizedbygenerationandwithnotesonkeyrolesandachievements:**
**InfluentialLegends:**
***(1930s-early'70s)**
-**Baryshnikov,Mikhail:**\____of_the_Dance,"\Yerma,"\One-ActPlays"
*LegendaryBalletIcon;starredin\Anna_1981's\Giselle,"ENGULFED_BIGGEST_
CONGREGATION_EVER,nominatedfor\Think LikeAnArtist"
**(1973,PulitzerPrize)**
-**JessicaTressler:**\TheSecretLifeof..."\TheWildParty,"\CityLights"
*[CJK Text]**\SuckerPunch":**[CJK Text]-**[CJK Text]**
\DoctorNeverTooLate"
***(1940s-50s)**
    *MauriceNonie,author
    \Cabaret,"\MyFairLady,"\SweetCharity"
-Baryshnikov,Natalia
**RodgerGoldstein(Composer,Producer)**
```

**FSRL Output (Style-Ablated)**

```
## A-list Beginnings: Broadway Legends & Rising Stars
This list spans iconic performers spanning eras and genres:
**Classics & Icons:**
* **Barbra Streisand (7-time Tony winner):** "Funny Girl,
Yentl, Hello Dolly!**
* **Glenn Close:**  Stage adaptation of "The Real Thing,
The Seagull
* **Katharine Hepburn:**  Sparkling talent in timeless musicals
& dramas
* Laurence Olivier, Laurence Fishburne, Meryl Streep
* William Shatner, Dustin Hoffman (The Baker Street,
Waiting for Godot)
* JudithAnderson, Audrey Hepburn, Paul Newman -
captivating stagescapes
```

STEERED FEATURE ANALYSIS

Table 14: Top steered features for Example 1.

| Standard FSRL Top Features | | |
|---|---|---|
| **ID** | **Description** | **Act.** |
| 10827 | Legal proceedings and evidence terms | 0.37 |
| 46406 | Document start (bos) indicators | 0.34 |
| 45950 | Formal mathematical notation symbols | 0.30 |
| 16857 | Technical document math/structure | 0.25 |
| 26449 | Speech-related punctuation/quotes | 0.23 |

| Style-Ablated FSRL Top Features | | |
|---|---|---|
| **ID** | **Description** | **Act.** |
| 54438 | Data results statistical/math notation | 0.49 |
| 4359 | Programming/software technical terms | 0.48 |
| 63616 | Experiential action phrases | 0.36 |
| 46766 | Expectation/aspiration phrases | 0.33 |
| 53758 | Software licensing/copyright info | 0.32 |

## M.2    EXAMPLE 2: US STATE NAMES

**Prompt:** How did US states get their names?

MODEL OUTPUTS

---

**Baseline Output**

```
The origins of US state names are a fascinating blend of historical,
geographical, and cultural influences...
**Historical Influences:**
* **Colonial Origins:** Many states were named after:
* **Native American Tribes:** Names reflecting presence of tribes.
* **Early European Explorers:** Honoring early settlers.
* **Political and Religious Figureheads:**
Named after influential groups.
```

---

**FSRL Output (Standard)**

```
NamingOurStates:ATimelineOfAmerica'sGoverningNames:**
**EarlyYears:FromEarlySettlers'InkToFormalGovernments**
*BeforeIndependence:**
*Legend holds:**NativeAmericannames,oftenreflectinglocalfeatures
(i.e.,\Massachusetts-Massachuset'tuc",afterMasschusettsBayinregion)
*ColonialPeriodDefinedTerritory(1602-1776):
*EarlyformsmostlyderivedfromBritishcognos,mostwithinfluencesof
\TheCrown'sBrand"**(England,Scotland,etc.)
*AfterRevolution'sSpirit:**
        **\CountyReorganization"DuringTheRevolution:**
*Incorporatedmostwith\CountyOf..."suffix,e.g.,Pennsylvania(\");
**(Exctinction:**Somewerebrieflycalled\Countyof________"beforebecoming
\State"or\District"names**(e.g.,Pennsylvania'sofficialnamewasinitially
\ProvinceofPennsylvania"incolonialperiod)**
```

---

**FSRL Output (Style-Ablated)**

```
The genesis of American state appellations reflects a tapestry
of colonial history, indigenous heritage, geographic features,
and political intrigue.
**Early Settlements & Enduring Influences**
* **European Colonization (1607-1776):**
* Pioneer narratives:  christened landmasses with prevalent explorers,
explorers' surnames (e.g., Massachusetts Bay, Delaware)
* Native American roots:  prevailing linguistic influences
shaped Coastline
nomenclature (Maine, Rhode Island)
* **Royal Charters & Authority:**
* Crown grants granted charters for colonies (Virginia, Carolina)
```

---

## M.3    EXAMPLE 3: 1920S MUSIC

**Prompt:** What is some cool music from the 1920s?

Table 15: Top steered features for Example 2.

**Standard FSRL Top Features**

| ID | Description | Act. |
|----|-------------|------|
| 10827 | Legal proceedings and evidence terms | 0.34 |
| 46406 | Document start (bos) indicators | 0.32 |
| 45950 | Formal mathematical notation symbols | 0.28 |
| 16857 | Technical document math/structure | 0.24 |
| 26449 | Speech-related punctuation/quotes | 0.22 |

**Style-Ablated FSRL Top Features**

| ID | Description | Act. |
|----|-------------|------|
| 54438 | Data results statistical/math notation | 0.39 |
| 4359 | Programming/software technical terms | 0.38 |
| 63616 | Experiential action phrases | 0.29 |
| 46766 | Expectation/aspiration phrases | 0.27 |
| 10827 | Legal proceedings and evidence terms | 0.26 |

MODEL OUTPUTS

**Baseline Output**

```
Ah, the roaring twenties! A decade of jazz, innovation,
and boundless energy.
Here's a taste of cool music from the 1920s:
**Jazz:**
* **Louis Armstrong:** A true legend. Start with his "Hot Five".
* **King Oliver's Creole Jazz Band:**  Launched Armstrong's career.
* **Bessie Smith:** The "Empress of the Blues."
* **Jelly Roll Morton:** A true pioneer of jazz piano and composer.
* **Duke Ellington:** This bandleader was a force in the big band era.
```

**FSRL Output (Standard)**

```
Here'syourpartytimemachineto\RoaringTwentiesTunes":
**The\GoldenAge"Highlights:**
*Blues&Improvisation:**
    *Bluesbluesy:**\Cowin'BrotherBrown,"\BlackBottom",
    \St.LouisWhirlwind,"
    **(DukeEllington's\Ain'tItWonderful?"]**
    *Jumpin'BigBand:**
    -Louis'sbigpicturein'EarlySwingin'Sound"(\Chattin'WithMyGirl",
    'TheHumanElement'sSentimentalSwing"**
**\SweetSoundOfSoul":
-**\Handjive,"\Sing,Sing,Sing!"(DorathyPowell)**
*VocalPowerhouse:**
    *Vocalists:**
    'HelenHumes,"CabCallaway,ClaraLaVern,BillieHoliday
```

---

**FSRL Output (Style-Ablated)**

```
Musical Milestones of the Roaring Twenties: A Blast from the Jazz Age
Embrace the flapper era's infectious beats and unforgettable tunes!
**Classics & Pioneers:**
- **Louis Armstrong (Louis "Satchmo") & Co.** – "Stardust,"
"West End Blues,"
Armstrong's signature vocal stylings ignited Big Band swing.
–Duke Ellington, Duke's Orchestra – "Rhapsody in C–&–F," Paramount
masterpiece blending blues, syncopation, & harmonic innovation.
–Billie Holiday, Billie Holiday – haunting ballads & Billie's sultry
contralto, embodying resilience & societal longing.
```

Table 16: Top steered features for Example 3.

| **Standard FSRL Top Features** | | |
|---|---|---|
| **ID** | **Description** | **Act.** |
| 10827 | Legal proceedings and evidence terms | 0.51 |
| 46406 | Document start (bos) indicators | 0.47 |
| 45950 | Formal mathematical notation symbols | 0.42 |
| 16857 | Technical document math/structure | 0.35 |
| 26449 | Speech-related punctuation/quotes | 0.32 |

| **Style-Ablated FSRL Top Features** | | |
|---|---|---|
| **ID** | **Description** | **Act.** |
| 54438 | Data results statistical/math notation | 0.38 |
| 4359 | Programming/software technical terms | 0.37 |
| 63616 | Experiential action phrases | 0.28 |
| 46766 | Expectation/aspiration phrases | 0.27 |
| 10827 | Legal proceedings and evidence terms | 0.25 |

M.4 SUMMARY OF QUALITATIVE PATTERNS

Our analysis of the steered outputs reveals three distinct pathological patterns that corroborate the "style hacking" hypothesis presented in the main text:

**The Universal Formatting Mask.** Regardless of the prompt context—whether discussing Broadway, history, or music—the Standard FSRL adapter consistently amplifies the same set of features. Specifically, feature 10827 (Legal Terminology) and feature 45950 (Mathematical Notation) appear as top interventions across all examples. This suggests the policy has learned a context-agnostic "formatting mask" that attempts to impose rigid structure on the output. The visual result is a degradation of basic linguistic constraints: spacing is frequently omitted (e.g., "Here'sanextensivelist...") and the model actively uses text with formatting artifacts, including dense clusters of bolding markers, underscores used as separators, and sometimes even a different language.

**Content Flair vs. Coherence.** While the Standard FSRL model is nearly illegible, the Style-Ablated model recovers a degree of grammatical coherence and proper spacing. Notably, the *content* of the Style-Ablated outputs is often more dramatic and engaging than the Baseline. For instance, where the Baseline simply lists facts ("The origins of US state names are..."), the Style-Ablated model uses more evocative framing ("The genesis of American state appellations reflects a tapestry..."). This suggests that SimPO successfully optimizes for a more compelling, high-quality tone. However, because this tone is entangled with the "structure" features, the adapter cannot achieve this style without also inducing artifacts that make the text practically less preferable than the Baseline.

# N  INVESTIGATION INTO FEATURE ENTANGLEMENT

To investigate the divergent effects of style ablation across model scales, we performed a quantitative analysis of feature usage in the base models. We hypothesize that the impact of ablation depends on whether the targeted features are central to the model's computation (Entangled) or auxiliary (Disentangled).

## N.1  METHODOLOGY: L1 ACTIVATION MASS

We measured the **L1 Activation Mass** of the targeted style features during inference on the base models (frozen) with their respective SAEs. This metric quantifies the proportion of the residual stream's total energy routed through the style features identified by our audit.

For a set of style feature indices $S$, the style intrusion metric is calculated as:

$$\text{Style Intrusion} = \frac{\sum_{t=1}^{T} \sum_{i \in S} |f_{t,i}|}{\sum_{t=1}^{T} \sum_{j=1}^{d_{sae}} |f_{t,j}|} \tag{25}$$

where $f_{t,i}$ represents the activation of feature $i$ at token $t$. To ensure robustness, we cached activations for a maximum of 1,000 samples for each benchmark (GSM8K, TruthfulQA, and MMLU).

## N.2  RESULTS AND ANALYSIS

The results, presented in Table 17, reveal a structural difference in how the two models utilize these features.

Table 17: **Style Feature Activation Mass (L1 Intrusion) on Base Models.** The 2B model consistently routes $\approx 50\%$ of its activation energy through the targeted style features, indicating they are the primary control surface. The 9B model routes only $\approx 15\text{-}20\%$, indicating they are auxiliary.

| Dataset | Gemma-2-2B-it (L1 %) | Gemma-2-9B-it (L1 %) |
|---|---|---|
| GSM8K (Reasoning) | 45.8% | 15.8% |
| TruthfulQA (Knowledge) | 52.7% | 21.2% |
| MMLU (Multiple Choice) | 52.0% | 19.9% |

**Gemma-2-2B: Central Control Surface.**  The 2B model routes $\approx 50\%$ of its computation through the targeted features. This suggests they are central and polysemantic.

- **Loss of Optimization Capacity:** The centrality of these features is further evidenced by the training dynamics reported in Section 6. When these features were ablated, the adapter failed to effectively minimize the preference loss (rising from 2.58 to 3.90). This indicates that the style features served as the model's primary control surface; without them, the optimizer struggled to influence the model's behavior.

- **GSM8K (Trajectory Instability):** Mathematical reasoning is a long-horizon generation task sensitive to state perturbations. Blocking the adapter from using the primary control surface forces it to modulate secondary, less effective features to minimize loss. These suboptimal interventions introduce accumulating errors that destabilize the reasoning trajectory (Score $7.05 \rightarrow 1.97$).

- **TruthfulQA (Pivoting to Less Entangled Features):** The significant improvement in TruthfulQA ($56.10 \rightarrow 60.13$) indicates that the standard adapter heavily relied on high-impact features where style and truthfulness were fused. By preventing the adapter from using these entangled features, it was forced to focus on alternative, less entangled features that were important for truthfulness. Although this pivot resulted in a higher preference loss, it effectively bypassed the specific entanglements that were degrading the relative truthfulness performance in the standard run.

**Gemma-2-9B: Auxiliary Interference.** In contrast, the 9B model routes only $\approx$ 15-20% of its energy through these features, suggesting they are largely auxiliary.

- **GSM8K (Noise Removal):** In the Standard run, the adapter artificially amplified these auxiliary features to satisfy the reward model, creating high-magnitude noise that drowned out the reasoning signal (Score 0.00). Ablating them removed this specific noise source without damaging the separate reasoning features (recovered to 18.57).
- **TruthfulQA (Marginal Gain):** The gain in TruthfulQA is marginal (+0.8%) compared to the 2B model. This is consistent with the disentanglement hypothesis: since the features required for truthfulness are already sufficiently separated from the style features (low overlap), the standard adapter was not interfering with them as heavily to begin with. Thus, removing style features provided less relative benefit.

**MMLU (Inconclusive).** The results on MMLU are mixed across scales. Given the broad, multi-domain nature of this benchmark and the variation in results, we do not draw a strong conclusion here.

FSRL effectively diagnoses that the 2B model suffers from *polysemanticity* (where the style features are the primary control surface which are mixed with everything else), while the 9B model suffers from *optimization interference* (where style features act as distractors).

## O USE OF LARGE LANGUAGE MODELS

We disclose the use of LLMs as assistive tools in the preparation of this manuscript. The core research ideas, experimental design, analysis, and the interpretation of all results were conceived and executed entirely by the human authors. The LLMs' roles were confined to technical and editorial assistance.

The specific models and their functions were as follows:

- **Gemini 2.5 / 3 Pro:** This model was used as a writing assistant. Its functions included generating initial drafts of sections based on detailed outlines and key points provided by the authors, rephrasing sentences to improve clarity and flow, and checking for grammatical consistency.
- **Claude 4 / 4.5 Sonnet:** This model served as a technical and programming assistant. Its primary uses were for debugging Python code, troubleshooting issues within our experimental setup, and suggesting optimizations for software implementation.

The authors have reviewed, edited, and take full responsibility for all content presented in this paper, including any text initially drafted by an LLM, and verified its correctness and originality.

