# OpenReview forum: "The Anatomy of Alignment: Decomposing Preference Optimization by Steering Sparse Features"
_ICLR.cc/2026/Conference — Submitted to ICLR 2026_

### Official Review · Reviewer_WRMW · 2025-10-19

**Soundness:** 3
**Presentation:** 2
**Contribution:** 2
**Rating:** 4
**Confidence:** 2

**Summary:**

The paper proposes Feature Steering with Reinforcement Learning (FSRL) — a framework that aligns large language models by steering one-layer interpretable sparse features rather than updating dense parameters directly. This allows for more transparent and auditable control over model behavior during preference optimization.

**Strengths:**

1. The proposed FSRL algorithm empirically optimizes preference alignment tasks using an interpretable basis, providing a possible interpretable framework for downstream task fine-tuning.
---
2. The authors provide causal analysis using zero-out steering on different feature categories and find evidence that preference alignment places most pressure on stylistic presentation rather than alignment-related concepts such as safety, honesty, and helpfulness.
---
3. Given the assumptions of local linearity and the empirical sparsity of active features, the authors build the theoretical relation between FSRL and the expressive power of LORA.  Specifically, the authors prove that the FSRL steering can be interpreted as an affine map given the local linearity assumption and relate FSRL’s soft-thresholding in sparse features to LoRA’s low-rank decomposition given the empirical sparsity assumption.

**Weaknesses:**

1. Requires more rigorous analysis of the trade-off between math reasoning degradation and preference alignment. It's unclear whether the preservation of math reasoning capability results from limited learnable capacity or from the authors' proposed methodology.
---
2. Table 3's intervention over feature activations is insightful, but more analysis is needed. What is the intervention loss difference between the unmodified model and the FSRL-trained model? Would the FSRL-trained model rely more on alignment features rather than disproportionately prioritizing style features? If not, what are the implications of observing this disproportionate prioritization of style-related over alignment-related features during preference alignment? The paper lacks empirical evidence supporting the claim that FSRL "provides a controlled framework for auditing alignment pressures".
---
3. The performance of FSRL is not sufficiently convincing (see Table 1). More empirical experiments are needed to demonstrate the benefits of FSRL—beyond interpretability—compared to interpretable model-diffing, e.g., other interpretable steering baselines like SAE-lens direct editing or neuron activation control.

**Questions:**

My questions are presented in the Weaknesses section.

---

> ### Author Response · Authors · 2025-11-21
>
> We thank the reviewer for recognizing the value of FSRL as a transparent framework for auditing preference optimization. We appreciate your request for deeper empirical analysis and comparisons.
>
> > **"It's unclear whether the preservation of math reasoning capability results from limited learnable capacity or from the authors' proposed methodology."**
>
> After correcting our SAE selection, we find that FSRL models do suffer from reasoning degradation (e.g., GSM8K drops from 53.45 to 7.05 on the 2B model). Our new causal experiments on the 9B model suggest this is due to **feature entanglement** within the SAEs rather than the adapter's capacity:
> *   **Gemma-2-9B-it:** When style features were masked (Style-Ablated), GSM8K performance recovered (0.00 to 18.57). This implies the specific SAE features utilized to optimize preference loss were entangled with math computations.
> *   **Gemma-2-2B-it:** Reasoning degraded further under ablation. This suggests the "backup" features the model pivoted to were even *more* entangled.
> This confirms FSRL effectively diagnoses specific causes of degradation, which vary by model scale.
>
> > **"What is the intervention loss difference... Would the FSRL-trained model rely more on alignment features rather than disproportionately prioritizing style features?"**
>
> With regards to the intervention loss between the unmodified and FSRL-trained model: We note that this is reported in Table 1, where the 2B Base model achieves a SimPO loss of 6.99 compared to the FSRL model's 2.58.
>
> Regarding feature reliance, our updated analysis using the corrected SAEs reveals a consistent trend: the adapter systematically amplifies style features while suppressing alignment features. However, we agree that observational trends are insufficient. To provide more robust empirical evidence, we conducted a new causal experiment: Style Ablation during Training.
>
> We trained new adapters where the style features identified by our audit were forced to zero. While the preference loss increased, the actual quality of the model improved in key areas. The Style-Ablated models achieved higher TruthfulQA scores and significantly better generation coherence on AlpacaEval compared to the standard FSRL models. This confirms that FSRL successfully audited the alignment pressure, revealing that the optimization process was prioritizing style at the expense of truthfulness and coherence.
>
>
> > **"More empirical experiments are needed to demonstrate the benefits of FSRL compared to other interpretable steering baselines like SAE-lens direct editing or neuron activation control."**
>
> Manually steering features is unscalable, as it requires knowing *a priori* which features to manipulate for complex behaviors. To provide a fair comparison with automated methods, we implemented two static steering baselines: **CAA and SAS**.
> *   These static methods fail to minimize the SimPO preference loss (~5.4 compared to FSRL’s ~2.6).
> This demonstrates that complex alignment tasks require the dynamic, token-dependent control provided by FSRL.
>
> > **"More empirical experiments are needed to demonstrate the benefits of FSRL compared to interpretable model-diffing."**
>
> Interpretable model-diffing is a nascent field requiring **Cross-Coders** to map features between base and fine-tuned models, as internal structure shifts during training. This is less scalable than our method, as it would require training a new heavy-weight Cross-Coder for every fine-tuned variation. FSRL avoids this by operating on a fixed, pre-trained SAE basis. Given these constraints, we believe such a comparison is best left for future work.

---

### Official Review · Reviewer_m73n · 2025-10-29

**Soundness:** 3
**Presentation:** 4
**Contribution:** 3
**Rating:** 6
**Confidence:** 3

**Summary:**

This paper introduces Feature Steering with Reinforcement Learning (FSRL), a framework for interpretable alignment of large language models. With pretrained sparse autoencoder (SAE) and frozen LLM, FSRL trains a lightweight adapter to modulate interpretable SAE features of a frozen LLM, rather than performing full-model fine-tuning. Additionaly, the authors theoretically establish that FSRL's activation-space corrections are equivalent to a restricted class of LoRA updates, inheriting LoRA's expressive power. Empirically, they apply FSRL to preference optimization on the UltraFeedback dataset using the Gemma-2-2B-it model and pretrained SAEs from GemmaScope, and demonstrate a moderate gain in MMLU, TruthfulQA, GSM8K compared to baseline while enabling interpretable interface. Through causal analysis based on their method, they find that style features contribute more than alignment features, suggesting preference optimization treats stylistic presentation as a proxy for quality rather than prioritizing deeper alignment concepts.

**Strengths:**

< Strength >

- The work addresses an important and timely problem. The opacity of standard alignment methods makes it difficult to diagnose issues like reward hacking. The use of SAE is theoretically justified and empirically shown for its interpretability.
- The theoretical justification connecting FSRL to LoRA updates is well-constructed, providing principled grounding for the approach. The limitation of the justification, the needs for adaptation across all layers, and empirical evidence is also discussed sufficiently.
- The paper is well-written with precise specification of the methodology. Figure 1 effectively illustrates the architecture.
- The causal ablation study provides unique compelling evidence for the disproportionate importance of style features, going beyond correlational analysis and has practical impact

**Weaknesses:**

< Weakness >

- The claimed advantage on interpretability still on a coarse high-level. Although the main advantage of the method is interpretability, the paper doesn’t have generation samples for steered generation except one in Figure 1. The fact that the policy was highly distributed (mentioned in the Appendix) implies the interpretability based on the proposed method remains on coarse level. As Figure 1 describes the methods as a sample-wise fine-grained steering, the paper should explicitly mention about the coverage in the main paper.
- Similarly, the analysis focuses on relatively simple, coarse-grained concepts of alignment vs. style, and MCC 0.448 for alignment features also raise concern for the substantial noise.
- Further, given the marginal performance gains in Table 1, this limited interpretability is difficult to justify.
- SimPO loss is used as the primary metric throughout (Tables 1 and 3), but this is merely the training objective, not a direct measure of alignment quality. Larger gap on SimPO loss between Base and FSRL (37.19%) in Table 1 despite a marginal gain on downstream performance (14.44% on MMLU and 0.71% on TruthfulQA) also raise the question on the performance. The addition of standard metrics like win-rates evaluations (even with LLM-as-judge) would bridge this gap.
- The degradation of full fine-tuning on mathematical reasoning (GSM8K) is acknowledged as a limitations from SimPO, however the paper doesn’t explain why FSRL shows significantly less degradation than full fine-tuning despite both optimizing the same SimPO objective.
- Is this due to insufficient training? The authors mention checking SimPO loss convergence would help
- DPO, a more popular and GSM8K-friendly baseline, was not evaluated despite acknowledged SimPO limitations

< Minor issues >

- $\boldsymbol{\theta}\in R_{+}^{d_{\text{sae}}}$ in adapter is confusing with $\theta$ in $\pi_\theta$, as those two models are different modules.
- Typo in line 105, “adding adding”

**Questions:**

<  Questions >

- Can you provide more qualitative generation examples showing how style and alignment feature steering manifests in actual text? While Figure 1 offers one illustration, more diverse qualitative samples would help readers better understand the behavioral changes induced by FSRL and strengthen the interpretability claims.
- In lines 349-351, “It enables researchers to causally link undesirable behaviors to the optimization of specific feature categories, thereby clarifying their root causes.”, could you provide empirical evidence demonstrating that steering specific feature categories can successfully mitigate or induce undesirable behaviors? The current results in Table 1 show that FSRL does not fully match the alignment efficacy of full fine-tuning, which makes this claim somewhat difficult to catch from the presented experiments.
- Would it be possible to supplement the evaluation with more standard alignment metrics such as win-rates or LLM-as-judge assessments? These metrics would provide a more direct measure of alignment quality.
- Given that DPO is more widely adopted in the community and is known to work well with mathematical reasoning tasks, could you provide results using DPO as the optimization objective? This would help clarify whether the findings such as style-over-alignment phenomenon or FSRL better performing on math are specific to SimPO or represents a more general characteristic of preference optimization methods.

---

> ### Author Response · Authors · 2025-11-21
>
> We thank the reviewer for their thorough evaluation. We have addressed your requests for qualitative samples, standard metrics, and empirical validation below.
>
> > **"The claimed advantage on interpretability still on a coarse high-level…"**
>
> We revised our architecture to enforce a non-negativity constraint, aligning FSRL with the with the SAE decoder's training assumptions. This drastically reduced steering vector density: the 9B model's adapter L0 (**58**) is now significantly sparser than the underlying SAE (**130**).
> This sparsity enables granular inspection. As shown in **Table 4**, the model targets specific formatting elements (e.g., em dashes, markup). This demonstrates that while we report aggregate statistics for clarity, the underlying steering operates on distinct, interpretable features that could enable more granular analysis.
>
> > **"Can you provide more qualitative generation examples...?"**
>
> We added **Appendix M** containing verbatim outputs. We find the FSRL adapter minimizes preference loss by forcing formatting markers (e.g., bolding) regardless of context. This provides qualitative confirmation of the style bias detected by our quantitative analysis.
>
> > **"Could you provide empirical evidence demonstrating that steering specific feature categories can successfully mitigate or induce undesirable behaviors... Would it be possible to supplement the evaluation with more standard alignment metrics such as win-rates or LLM-as-judge assessments?"**
>
> We evaluated new "Style-Ablated" adapters (style features masked) on **AlpacaEval 2.0**:
> - **Standard FSRL:** The model achieves a low win rate (<1%), indicating that the optimization of the preference objective led to a loss of coherence.
> - **Style-Ablated FSRL:** When style features are ablated, the win rate improves significantly (e.g., from 0.20% to 5.57% on the 9B model), and the model recovers part of its generation coherence.
> - **Conclusion:** This confirms our diagnostic finding. The incoherence was caused by the model over-optimizing specific style features. By surgically removing them, we were able to partially restore the model's capability.
>
> > **"The paper doesn't explain why FSRL shows significantly less degradation... Is this due to insufficient training?"**
>
> On whether the degradation in reasoning could be due to insufficient training: We have included training and validation loss curves in Appendix D (Figure 3). These plots demonstrate that the models have fully converged and do not exhibit overfitting, ruling out insufficient training as the cause.
> Instead, we observe interesting divergences when style features are ablated.
> - **Gemma-2-9B-it:** Reasoning performance recovers significantly (from 0.00 to 18.57) when style features are ablated. This implies that the primary style features targeted by the standard policy were interfering with reasoning computations.
> - **Gemma-2-2B-it:** Reasoning performance drops further (from 7.05 to 1.97) under ablation. This suggests that when the primary style features were removed, the adapter pivoted to alternative features to optimize the loss, and these secondary features were even more entangled with mathematical reasoning.
>
> > **"The analysis focuses on relatively simple, coarse-grained concepts... MCC 0.448... raise concern for the substantial noise."**
>
> To address classifier noise (MCC 0.448), **Appendix L** details a sensitivity analysis using precision scores from our manual validation. We calculated a **Lower Bound Ratio** for causal impact, assuming a worst-case scenario where all false positives contribute zero impact. Even under these assumptions, style features remain significantly more impactful than alignment features.
>
> > **"Could you provide results using DPO as the optimization objective?"**
>
> Our implementation uses analysis-focused libraries (TransformerLens/SAE-Lens), creating memory bottlenecks (max batch size 2 for 2B). DPO requires a reference model, doubling memory overhead, which was infeasible given our constraints. Theoretically, both DPO and SimPO optimize the same underlying Bradley-Terry reward model. The primary difference is that DPO includes an implicit KL divergence penalty, which mitigates the drift into repetitive stylistic patterns that we observe with SimPO.

---

### Official Review · Reviewer_bEBw · 2025-10-29

**Soundness:** 3
**Presentation:** 3
**Contribution:** 2
**Rating:** 4
**Confidence:** 4

**Summary:**

This paper introduces Feature Steering with Reinforcement Learning (FSRL), where an adapter is learned from preference data to steer model behavior. In particular, the adapter operates over the activations of a particular layer in the model. The adapter learns to perturb these activations by modulating additive features learned by SAEs that correspond to higher level semantic concepts (e.g., alignment or style). This results in a method that can steer model behavior by only updating the activations for a single layer in a model in a supposedly more interpretable manner.

The authors show that FSRL improves the preference loss of a 2B model on several standard benchmarks, comparing to SimPO. The authors then also show that the proportion of features corresponding the “alignment’ and “style” concepts decreases when training with FSRL, and that preventing adaptations to certain features (either “alignment” or “style”) impacts the model’s resulting loss differently. These results indicate that certain high-level features—in this case “style”—are more important for reducing preference loss than others.

Finally, the authors also prove that FSRL updates fall within the class of LORA updates, providing theoretical justification for their method.

**Strengths:**

The author’s theoretical analysis makes an interesting connection between LORA updates and steering model behavior via updating only the activations of the model. Additionally, the paper is well written and easy to follow. The method, FSRL, is clearly explained and well situated in prior work. FSRL is also evaluated on standard benchmarks making it easy to understand comparisons to prior work.

**Weaknesses:**

I have two major concerns: (1) I am left unsure of the benefit of FSRL over full-fine tuning (2) I am not convinced of the novelty of this work. I additionally have a third, lower priority concern (3) the mechanistic insights could use more explanations or analysis.

Regarding (1): Table 1 shows that SimPO outperforms FSRL across all benchmarks except for GSM8K, but here FSRL is outperformed by the base model. In terms of performance, it is not clear to me that there is any benefit to using FSRL; I don’t think it is sufficient to argue that FSRL is more performant than SimPO on one benchmark (although worse than the base model on that benchmark) but less performant on all others. Additionally, the authors of SimPO indicate that their method may do worse on GSM8K because of the training data; if you use different training data (e.g., perhaps more tailored to reasoning tasks) do you still see relatively better performance than SimPO? Its also not clear to me if FSRL possesses a benefit in terms of interpretability: the authors note that the FSRL adapter outputs a steering vector that is significantly more dense than the SAE’s features, and the empirical evaluations with regards to interpretability are limited. Finally, it is not clear if FSRL possesses any benefits in terms of computational expense: while it only requires limited adaptations to the models activations for a single layer, finding that layer required extensive search. Therefore, I am left unsure of where FSRL outperforms prior work.

Regarding (2): This recent work from Bayat et al. (https://arxiv.org/pdf/2503.00177) also focuses on learning to steer a model by leveraging SAEs with targeted updates. Bayat et al. also learns how to steer model behavior from preferences, and evaluates on the same benchmarks as this submission. I think this work would be a fair comparison empirically—-or at least the authors should justify conceptually how their work differs.

Regarding (3): The authors note that, given Table 2, “the adapter learns to decrease the proportional activation of both alignment and style features relative to the baseline.” Why would the adapter learn to decrease the number of alignment features and style features? What type of behavior does this correspond to? My initial thought is that this observation might indicate that the adapter results in different features that are not classified as alignment or style features, possibly because these features are now different from the ones used to train the classifier. These empirical results do not tell me sufficient information about how the model behavior is changing. This analysis, without further explanation or connection to model behavior, feels incomplete.

**Questions:**

Where is the benefit of FSRL over prior work?
What is the novelty of FSRL with respect to Bayat et al.? Why do you observe decreasing proportions of alignment and style features after training with FSRL?

---

> ### Author Response · Authors · 2025-11-21
>
> We thank the reviewer for their detailed feedback and for identifying the connection to recent work on sparse steering. Below, we address your concerns regarding the utility, novelty, and computational cost of FSRL with new experiments and analyses.
>
> > **"I am left unsure of the benefit of FSRL over full-fine tuning. Table 1 shows that SimPO outperforms FSRL across all benchmarks except for GSM8K."**
>
> You correctly noted that FSRL does not outperform full fine-tuning on standard benchmarks. In our revised manuscript, we have clarified that **FSRL is not intended to replace full fine-tuning as a production alignment tool.** Instead, it acts as a transparent diagnostic instrument. Its value lies in dissecting the alignment process to reveal how the model minimizes the preference loss, even when that optimization degrades performance.
>
> To demonstrate this utility, we added a causal ablation study in **Section 6**. By training new adapters with the identified style features masked, we found that the model achieved significantly higher scores on TruthfulQA and recovered generation coherence compared to the standard FSRL model. This confirms that FSRL successfully diagnosed the root cause of the degradation: the optimization process was prioritizing stylistic features at the expense of truthfulness. This kind of mechanistic audit is not possible with opaque full fine-tuning.
>
> > **"Its also not clear to me if FSRL possesses a benefit in terms of interpretability: the authors note that the FSRL adapter outputs a steering vector that is significantly more dense than the SAE's features."**
>
> Regarding your concern about the density of the steering vector, we have improved our architecture by enforcing a non-negativity constraint on the steered features to align with the SAE decoder's training assumptions. This drastically reduced the density of our steering vectors compared to our previous submission (from ~360 down to 95 for the 2B model).
>
> *   For the 2B model, the adapter's L0 (95) is comparable to the base SAE's L0 (73).
> *   For the 9B model, the adapter is significantly sparser (L0 of 58) than the base SAE (L0 of 130).
>
> This confirms that the learned policy operates within a sparsity regime that is comparable to, or even more interpretable than, the underlying SAE activations.
>
> Regarding the mechanistic insights in **Table 2**, we identified an oversight in our previous SAE selection and explanations from Neuronpedia. After retraining with the correct canonical SAEs, the trends in feature proportions have become clearer. We now observe a consistent (and more prominent) pattern across both 2B and 9B models where the adapter suppresses alignment features while significantly amplifying style features.
>
> > **"Finally, it is not clear if FSRL possesses any benefits in terms of computational expense: while it only requires limited adaptations to the models activations for a single layer, finding that layer required extensive search."**
>
> We addressed the concern regarding the cost of finding the intervention layer in **Appendix C**. We trained simple logistic regression probes to distinguish between chosen and rejected responses across model layers. We found that the accuracy of these linear probes peaks at Layer 12 (54.75%), the same layer identified by our expensive full training sweeps. This confirms that the optimal intervention layer can be identified using computationally cheap linear probing rather than resource-intensive sweeps.
>
> > **"This recent work from Bayat et al. also focuses on learning to steer a model by leveraging SAEs with targeted updates. I think this work would be a fair comparison empirically."**
>
> We agree that Bayat et al. is a relevant comparison. However, a fundamental distinction exists between the methods. SAS derives a static steering vector from contrastive pairs, applying the same intervention to every token. FSRL learns a dynamic, context-dependent policy via an adapter that can be optimized against any differentiable objective.
>
> To quantify this difference, we implemented SAS as a baseline in **Appendix H**. We found that the static SAS vector failed to significantly minimize the SimPO preference loss (achieving a loss of ~5.4 compared to FSRL’s ~2.6). This demonstrates that complex preference optimization requires the dynamic, token-by-token control provided by our adapter, differentiating our contribution from static steering approaches.

---

> > ### Comment · Reviewer_bEBw · 2025-11-25
> >
> > Regarding your additions to Section 6; I am admittedly not convinced that you have demonstrated the utility of your method  as a "transparent diagnostic instrument". You cite that by masking style features, your model achieved significantly higher scores on TruthfulQA compared to the standard FSRL model. But that difference is only present for the 2B model, not the 9B model; if the utility of this method is in its ability to serve as a transparent diagnostic instrument then why do these results not hold for a larger model? Unless I am missing the bigger picture, there doesn't seem to be a clear pattern in the results in Table 5 regarding standard vs. style-ablated FSRL that holds across both models; in 3/5 of the datasets you consider, the best version (standard or style-ablated FSRL) switches depending on the model size, and in the other 2/5 datasets the performance between each version is very close for the 9B model.
> >
> > I am more convinced that FSRL's steering vectors are within the standard interpretability regime for current SAEs, thanks for clarifying that! Also, thanks for comparing against Bayat et al.
> >
> > All this being said, I do not have sufficient evidence to raise my score. If the primary contribution of this work is in improved interpretability, then this needs to be analyzed more explicitly. I don't think producing steering vectors with the same L0 as SAE's is sufficient evidence that this method produces meaningfully interpretable steering vectors. The insight that Table 2 now shows is interesting; the adapter learns to decrease the proportion of alignment features and increase the proportion of style features. Can the authors show this trend for other types of features when training with a different dataset? This paper only presents a single instance of a useful mechanistic insight, but that feels insufficient to claim that the method is indeed a robust contribution to interpretability research. Re-doing the analysis that produced Table 2 for other settings would greatly strengthen the paper. Additionally, can the authors further discuss/investigate the seemingly inconsistent results in Table 5? This would help answer the question  "how can we use the interpretable steering vectors learned by FSRL for training?"

---

> ### Author Response · Authors · 2025-11-25
>
> We thank the reviewer for their continued engagement. We address your remaining concern regarding the divergent results between the 2B and 9B models and the request for additional datasets below.
>
> > You cite that by masking style features, your model achieved significantly higher scores on TruthfulQA compared to the standard FSRL model. But that difference is only present for the 2B model, … why do these results not hold for a larger model?
>
> You asked why the TruthfulQA (TQA) improvement is more noticeable in the 2B model. We argue that this divergence does not indicate randomness. Instead, it reflects a structural difference in how the underlying SAEs encode "style," which we have exposed. We have added **Appendix N**, which provides an in-depth hypothesis for the observed performance changes on each benchmark based on **L1 Activation Mass**.
>
> *   **Gemma-2-2B (Entangled Control):** We found that the targeted style features account for **~50%** of the residual stream's activation mass. They are the primary control surface. Because they are central and polysemantic, the adapter relies on them heavily. Ablating them removes this core pathway. While forcing a pivot to better truthfulness features improved TQA (**56.10** to **60.13**), it destabilized the reasoning trajectory required for GSM8K (**7.05** to **1.97**), as the model lost the stability required for generation that is provided by these central features.
> *   **Gemma-2-9B (Auxiliary Noise):** The style features account for only **~15-20%** of the mass. They are auxiliary. Regardless, the adapter uses them to satisfy the loss. This creates high-magnitude noise that breaks fragile reasoning circuits (GSM8K drops to **0.0**). Ablating them removes this noise and recovers GSM8K to **18.57**. It also yields a TQA improvement of **0.8**. This gain is smaller than in the 2B model because the features are less entangled to begin with.
>
> Crucially, FSRL succeeds as a diagnostic tool in both cases. In both models, preventing the adapter from using these features increased the SimPO preference loss (**Table 5**) yet improved generation quality and truthfulness. This provides mechanistic evidence of **Goodhart’s Law**: FSRL correctly diagnosed that the preference optimization process was prioritizing style over substance, regardless of how that pressure manifested in the specific SAE and model.
>
> > If the primary contribution of this work is in improved interpretability, then this needs to be analyzed more explicitly. … This paper only presents a single instance of a useful mechanistic insight, but that feels insufficient to claim that the method is indeed a robust contribution to interpretability research.
>
> Our core methodological contribution is that we explicitly designed FSRL to mirror the standard RLHF pipeline, distinguishing it from prior steering methods. While traditional steering vectors (like Bayat et al.) must be created for individual tasks separately (e.g., deriving a specific "honesty" vector to test on honesty data), FSRL trains on a general preference dataset (**UltraFeedback**), and we observe the downstream effects on distinct, unseen benchmarks (MMLU, GSM8K, TruthfulQA).
>
> Regarding the interpretability findings, we prioritized **depth of verification** over **breadth of datasets**. The "Style over Substance" phenomenon is an important finding with implications for the field since UltraFeedback is a dataset widely used in both research and industry for aligning models. Because the implications are severe, suggesting that blindly optimizing the preference loss on standard datasets harms intended alignment, we focused our analysis on rigorously verifying this single pathology.
>
> We emphasize that **Tables 2, 3, and 6 are mutually consistent** and collectively validate this insight:
> *   **Table 2 (Observation):** Shows a significant increase in firing rate of style features over alignment ones.
> *   **Table 3 (Mechanism):** Proves via causal analysis that style features are substantially more important for minimizing loss than alignment features.
> *   **Table 6 (Validation):** Confirms via AlpacaEval that this optimization actively degrades generation coherence.
>
> We believe that deeply auditing this specific pathology provides more value to the community than a shallow demonstration of feature comparisons across multiple datasets. The fact that FSRL could isolate this behavior and allow us to surgically intervene (improving TQA and win rates by ignoring the objective) demonstrates the utility of the framework.

---

### Official Review · Reviewer_T5Np · 2025-11-01

**Soundness:** 3
**Presentation:** 3
**Contribution:** 3
**Rating:** 4
**Confidence:** 3

**Summary:**

The paper presents a method (FSRL) for preference alignment with a lightweight adaptor that also enables interpretable steering. The method uses the SimPO objective and an adaptor to transform activations into features to pass into a sparse autoencoder for updating the residual stream. The authors show that FSRL's updates are equivalent to an input-dependent LoRA update, conduct some ablations on the form of the adaptor itself, demonstrate that the method yields performance between that of full finetuning and the original base model, and show that the trained adaptor can be used to provide insights into the alignment process (through observation and intervention on the latent features).

**Strengths:**

1. The paper is well-written: Figure 1 makes the high-level idea clear and experimental setups are well-explained.
2. The paper performs ablations to motivate the setup.
3. The paper provides an interesting analysis on the contribution of different high-level categories (e.g., alignment, style) to the post-training objective.

**Weaknesses:**

1. The paper could benefit from more direct comparisons with related work, including other forms of steering as well as other adaptors (such as non-interpretable ones). The latter are not discussed in the related work, and additional experimental comparisons would strengthen the paper in the context of the existing landscape of training with adaptors and interpretable steering. For instance, how does the additional interpretability of this setup compare to vanilla adaptors for training?
2. The paper only runs experiments on a single model. Running experiments on one additional model would strengthen the paper (and potentially offer important practical insights on the application of this method). For instance, can this method with the same SAE be used on a model from a different model family?
3. To be able to more confidently draw conclusions from section 5, it would be helpful to propagate the error from the categorization process into the results of the analysis. Do the claims still hold when this is taken into account?

**Questions:**

Please see the above for questions!

---

> ### Author Response · Authors · 2025-11-21
>
> We thank the reviewer for their positive assessment of the paper’s clarity and for highlighting the novelty of our analysis. Below, we address your specific concerns with new experiments and analyses included in the revised PDF.
>
> > **"The paper could benefit from more direct comparisons with related work, including other forms of steering as well as other adaptors (such as non-interpretable ones)..."**
>
> We agree that the paper benefits from a more direct comparison with existing methods. We have addressed this in three ways:
>
> *   **Taxonomy of Methods (Section 8):** We added **Table 7** to categorize methods based on their Target Space and Adaptivity. In this table, we explicitly situate FSRL against static steering and dynamic adapters.
> *   **Static Baselines (Appendix H):** We implemented two static steering baselines, CAA and SAS. Our results show that these static methods fail to minimize the SimPO preference loss (achieving a loss of ~5.4 compared to FSRL’s ~2.6). This demonstrates the necessity of the dynamic control provided by the FSRL adapter.
> *   **Comparison to Vanilla Adapters:** Standard adapters are designed to approximate full fine-tuning; thus, we do not include them as a separate baseline. The key distinction of FSRL is the intervention basis. Vanilla adapters operate in parameter space where superposition is prevalent, meaning a single neuron often encodes multiple unrelated concepts, making the update opaque. In contrast, FSRL operates on a sparse, interpretable feature basis, enforced by an L1 penalty on the adapter's activations to encourage sparsity. This constraint is absent in standard adapter training.
>
> > **"The paper only runs experiments on a single model. Running experiments on one additional model would strengthen the paper."**
>
> We clarify that SAEs are learned dictionaries specific to a model's internal activation space and cannot be transferred between families. However, to address the concern regarding generality, we replicated our experiments on **Gemma-2-9B-it**:
>
> *   **Results:** As detailed in **Section 5**, the core finding holds. The 9B model also minimizes loss by disproportionately leveraging style features over alignment features.
> *   **Implication:** The consistency of results across 2B and 9B scales suggests our findings regarding the prioritization of style are not artifacts of a specific model size.
>
> > **"To be able to more confidently draw conclusions from section 5, it would be helpful to propagate the error from the categorization process into the results of the analysis."**
>
> We thank the reviewer for rightly pointing out that the classification errors could affect our downstream causal analysis. To ensure that this isn’t the case, we performed a sensitivity analysis detailed in **Appendix L**.
>
> 1.  We used the precision scores derived directly from the confusion matrix of our manual validation to correct our estimates.
> 2.  We then assume a worst-case scenario where all false positive features contribute zero impact, and calculate a **Lower Bound Ratio** for the causal impact of features.
> 3. Even under these strict assumptions, the style category remains significantly more impactful than the alignment category. This confirms that our conclusion is robust to the noise inherent in LLM-based feature classification.

---

### Author Response · Authors · 2025-11-21
**Summary of our Changes**

We thank the reviewers for their insightful and constructive feedback. In response to the reviews, we have performed a major revision of the paper, including scaling our experiments to the Gemma 2 9B model, adding new baselines (CAA, SAS), and conducting extensive causal analysis.

Below is a summary of the key updates and new experimental results:
1. **Correction of SAE Selection and Retraining:** We identified an oversight in our initial submission where the SAEs used did not align with the feature explanations available on Neuronpedia. We have corrected this by selecting the appropriate canonical SAEs and retraining all adapters reported in the paper. Additionally, we introduced a non-negativity constraint (see Appendix A) on the steered features to ensure adherence to the SAE decoder's training assumptions. This architectural change significantly reduced the average L0 norm of the steering vector. While this updated training changed the magnitude of certain results, most notably the 2B model now exhibits a more dramatic drop in GSM8K performance, our core qualitative claims and findings remain constant. We have added a discussion in Section 6 explaining that this reasoning degradation likely stems from feature entanglement within the specific SAEs used.
2. **Reframing FSRL as a Diagnostic Instrument:** Reviewers bEBw and m73n noted that FSRL does not outperform full fine-tuning on standard benchmarks. While we did not originally intend to frame FSRL as a competitor to state-of-the-art fine-tuning, we acknowledge that the text may have implied this. We have revised the manuscript to explicitly position FSRL as a transparent diagnostic instrument. Its primary value lies in decomposing the opaque alignment process into interpretable feature modulations, allowing us to audit how the model satisfies the preference objective even when that optimization comes at the cost of generation coherence.
3. **Scaling to Gemma-2-9B-it:** To address concerns regarding the generality of our findings (Reviewers T5Np, m73n), we replicated our experiments on Gemma-2-9B-it. We find that:
- The "Style over Alignment" phenomenon holds across model scales.
- Mathematical reasoning (GSM8K) collapses in the 9B model, while improving in the style-ablated variants. This suggests that feature entanglement varies across model scales and depends on the specific underlying SAEs used.
- This confirms that FSRL provides consistent mechanistic insights across model sizes.
4. **Causal Validation via Style Ablation:** To causally link the style over alignment bias to model degradation (Reviewer m73n), we trained new adapters where style features were masked (ablated) during the training process.
- Result: These "Style-Ablated" models achieve significantly higher scores on TruthfulQA and recover generation coherence compared to the standard FSRL models.
- Implication: This confirms that the standard optimization process minimizes loss by prioritizing style features at the expense of semantic quality and truthfulness.
5. **Quantifying Coherence: AlpacaEval and Qualitative Analysis:** We introduced length-controlled AlpacaEval 2.0 win rates and a qualitative analysis of generated text (Appendix M, Reviewer m73n).
- Finding: Standard FSRL models suffer from a collapse in coherence (win rates < 1%), driven by an over-optimization of style features where the model forces artifacts, such as excessive bolding, regardless of the prompt.
- Mechanism: This highlights a failure mode of SimPO (which lacks a hard KL penalty) that FSRL makes visible. The model maximizes reward by driving specific stylistic features to extreme values, breaking coherence.
6. **New Baselines: Static vs. Dynamic Steering:** We compared FSRL against static steering methods: CAA and SAS (Reviewers T5Np, bEBw, WRMW).
- Result: Static methods fail to minimize the SimPO loss (reaching ~5.4 compared to FSRL’s ~2.6).
- Conclusion: Alignment is a complex, context-dependent task. A single static vector is insufficient. An input-dependent adapter (FSRL) is required to effectively optimize the preference objective.
7. **Robustness and Methodology**
- Sensitivity Analysis: We performed a sensitivity analysis (Appendix L, Reviewer m73n, T5Np) incorporating the confusion matrix of our automated classifier. Even in the worst-case scenario, the finding that the policy disproportionately targets style features remains robust.
- Computationally Cheap Layer Selection: We validated our intervention layer choice by training linear probes (logistic regression) across model layers. We found that Layer 12 achieves the highest classification accuracy (54.75%) for separating chosen vs. rejected responses. This confirms that the optimal intervention layer can be identified using computationally cheaper methods rather than expensive full training sweeps (Reviewers bEBw).

---

### Meta-Review · Area_Chair_ymWJ · 2026-01-05

**Summary:**

Reviewers found the paper well-motivated and theoretical results interesting, but raised concerns on insufficient experiments, insufficient comparisons with related work, and the unclear novelty.

**Reviewer Concerns:**

see above

**Reviewer Scores:**

NA

---

### Decision · Program_Chairs · 2026-01-26

Reject